# Structural characterization of the oligomerization of full-length Hantaan virus polymerase into symmetric dimers and hexamers

Quentin Durieux Trouilleton [1], Dominique Housset[1], Paco Tarillon [1], Benoît Arragain [1,3] ✉ & Hélène Malet [1,2] ✉

Hantaan virus is a dangerous human pathogen whose segmented negative-stranded RNA genome is replicated and transcribed by a virally-encoded multifunctional polymerase. Here we describe the complete cryo-electron microscopy structure of Hantaan virus polymerase in several oligomeric forms. Apo polymerase protomers can adopt two drastically different conformations, which assemble into two distinct symmetric homodimers, that can themselves gather to form hexamers. Polymerase dimerization induces the stabilization of most polymerase domains, including the C-terminal domain that contributes the most to dimer's interface, along with a lariat region that participates to the polymerase steadying. Binding to viral RNA induces significant conformational changes resulting in symmetric oligomer disruption and polymerase activation, suggesting the possible involvement of apo multimers as protecting systems that would stabilize the otherwise flexible C-terminal domains. Overall, these results provide insights into the multimerization capability of Hantavirus polymerase and may help to define antiviral compounds to counteract these life-threatening viruses.

*Bunyavirales* is a large order of segmented negative-stranded RNA viruses (sNSV) that encompasses several highly pathogenic zoonotic viruses[1]. Most Bunyaviruses are transmitted by mosquitoes or ticks, except Hantaviruses that are rodent-borne[2]. Human infection by Hantaviruses can result in severe diseases such as hemorrhagic fever with renal syndrome in the case of old-world Hantaviruses or encephalitis in the case of new-world hantaviruses[3]. Representative members of the old-world hantaviruses are Hantaan and Puumala viruses, whose infections result in fatalities in up to 15% of the cases[4], whereas new-world Andes and Sin Nombre (SNV) hantavirus infections have mortality rates of up to 40%[5]. Despite their potential threat to human health, neither drugs nor vaccines are presently available to combat or prevent infection by these types of viruses.

In this context, we focused our research on Hantaan virus polymerase (HTNV-L) that could be a key target for antiviral drug development. It catalyzes two fundamental steps of the viral cycle: replication that results in the duplication of the viral genome, and transcription that generates viral messenger RNA (mRNA). As for other Bunyaviruses, the replication process uses an internal prime-and-realign mechanism for its initiation in the absence of a primer[6–8]. Conversely, transcription is performed by host-cell mRNA cap-snatching. This intricate process involves the binding of host-cell mRNA through the polymerase cap-binding domain (CBD) and their subsequent cleavage after 10 to 16 nucleotides by HTNV-L endonuclease (ENDO), thus generating capped RNA primers used by the polymerase core to elongate the viral mRNA[6,7].

[1]Université Grenoble Alpes, CEA, CNRS, IBS, F-38000 Grenoble, France. [2]Institut Universitaire de France (IUF), Paris, France. [3]Present address: European Molecular Biology Laboratory (EMBL), Grenoble, France. ✉e-mail: arragain@embl.fr; helene.malet@ibs.fr

Knowledge of the *Bunyavirales* polymerase organization comes from several structures that have been determined in the last few years. These include the structures of La Crosse virus (LACV, *Peribunyaviridae* family)[9–11], Lassa and Machupo viruses (LASV and MACV, *Arenaviridae* family)[12–15], Dabie Bandavirus (DBV, previously named Severe Fever with Thrombocytopenia Disease virus, *Phenuiviridae* family), and Rift Valley Fever virus (RVFV, *Phenuiviridae* family). In the case of HTNV-L, its first structural characterization came from the isolated ENDO domain that was determined by X-ray crystallography[16]. More recently, cryo-electron microscopy structures of HTNV-L and SNV-L have unveiled the organization of Hantavirus polymerase core[8,17]. The interior of the core adopts a right-hand organization that is common to all RNA-dependent RNA polymerases (RdRp) and comprises a palm, a finger, and a thumb domains. This central part of the core is encircled by (i) the linker region that connects the ENDO to the core, (ii) the core-lobe that contains a vRNA-binding lobe (vRBL), (iii) the thumb-ring that surrounds the thumb, and (iv) the lid that closes the active site cavity. The active site cavity is located in the core center at the intersection of four tunnels: the RNA template entry, the nucleotide entry, the template exit, and the product exit tunnels[7,9]. The active site cavity contains the canonical catalytic motifs A to F that are conserved in all RdRp[18]. HTNV-L and SNV-L core structures were solved in an inactive configuration, wherein several of the catalytic motifs were found to be misplaced. In particular, the motif E, also known as "primer-gripper" which typically adopts a canonical three-stranded β-sheet conformation, was shown to display an unusual α-helical conformation in the absence of RNA. The binding of the 5'vRNA end as a hook in a specific binding site on the HTNV-L core was shown to induce a conformational rearrangement of the motif E into the more standard 3-stranded β-sheet structure, inducing several reconfigurations, notably the ordering of the putative prime-and-realign loop (PR loop) that might be involved in prime-and-realign initiation. Binding to the 5' vRNA end also induces the organization of motif F that is flexible in the apo form. In both HTNV-L and SNV-L structures, only the cores are visible, as flexibility prevents the visualization of the ENDO and of all the C-terminal regions.

Here, we present near-atomic resolution cryo-EM structures of the complete apo HTNV-L in various oligomeric states. The structures unveil the location of every domain of this central enzyme, notably depicting the organization of its C-terminal region that was uncharacterized. The flexibility of the polymerase is revealed, which results in the presence of two very distinct conformations of protomers, that can assemble into two singular symmetric homo-dimers. Dimerization implies a domain-swapping mechanism that could be conserved amongst Bunyaviruses. Our results reveal that the symmetric apo homo-dimers can further assemble into a hexamer composed of a trimer of dimers, forming a unique intricate organization that had never been observed in any polymerase. We show that 5'vRNA binding disrupts the equilibrium between monomers, symmetric dimers and hexamers, suggesting a role of the symmetric multimers as possible storage systems that would stabilize and protect apo HTNV-L. Our results finally reveal that incubation of HTNV-L with 5'vRNA triggers large conformational changes in the protomers leading to an equilibrium between active monomers and yet-to-be characterized 5'vRNA-related dimers, which are different from the apo dimers.

## Results

### Structures of full-length HTNV-L monomer, symmetric dimer, and hexamer

A full-length wild-type (WT) HTNV-L construct comprising an N-terminal his-tag was expressed in Hi5 insect cells and subsequently purified (Supplementary Fig. 1a, b). Through a combination of size-exclusion chromatography (SEC) and mass-photometry analyses, the presence of distinct HTNV-L oligomers was observed. In buffers containing 250 mM NaCl, several species were automatically detected

including monomers (53%), dimers (17%), higher molecular weight oligomers up to hexamers (2%) and an unidentified 150 kDa protein (18%) (Fig. 1a). Increasing the ionic strength to 1 M NaCl maintained the proportion of monomers/dimers, while no automatic detection was obtained for the oligomers ranging from trimers to hexamers (Supplementary Fig. 1c). Inversely, in a buffer containing 150 mM NaCl, a large proportion of the 150 kDa species were visualized (45%), while less monomers and dimers were present and higher oligomers were not detected (Supplementary Fig. 1c). To characterize the proteins present in the 150 kDa band, a bottom-up mass spectrometry (MS)-based proteomic analysis was performed. For this, proteins present in the 150 kDa band were in-gel digested with trypsin before analysis by nanoliquid chromatography coupled to MS/MS. The obtained result revealed that the most abundant protein in this band was HTNV-L (Supplementary Fig. 2a). The identified peptides were covering the entire HTNV-L sequence (Supplementary Fig. 2b), suggesting that the 150 kDa band may contain different cleaved forms of HTNV-L co-migrating in this particular SDS-PAGE gel band.

To further characterize the different full-length HTNV-L oligomers, we performed SEC coupled to mass photometry in a buffer containing 250 mM NaCl. The analysis of the different SEC fractions suggested that the various HTNV-L oligomers could be in equilibrium (Supplementary Fig. 1d). Size-exclusion chromatography coupled to static light scattering (SEC-SLS) confirms this equilibrium with the detection of a single peak whose weight-averaged molar mass increases from 296 to 374 kDa upon HTNV-L concentration increase from 2.4 to 14 μM (Fig. 1b). The complex mixture present in this single peak unfortunately prevents the deduction of the multiple dissociation constants between the different species.

A cryo-EM dataset was collected to determine the structures of the different species despite the observed equilibrium that hinders further biochemical separation. Particle picking was performed using both template and neural-network-based picker[19,20], thereby facilitating the selection and separation of monomers from both symmetric dimers and hexamers (Supplementary Figs. 3, 4). From this dataset, multiple structures of apo HTNV-L were obtained in different oligomeric states: (i) the structure of monomeric apo HTNV-L WT, determined at an overall resolution of 2.6 Å, exhibits a conserved architecture compared to the previously obtained apo HTNV-L$_{D97A}$[8] (Fig. 1c, Supplementary Fig. 3, and Supplementary Table 1). The polymerase core is clearly visible, whereas the ENDO and the C-terminal region are too flexible to be resolved. (ii) The structure of an isolated apo HTNV-L WT symmetric dimer, that is determined at an overall resolution of 3.0 Å resolution, reveals the location of all domains, including the ENDO and the C-terminal region (Fig. 1d and Supplementary Fig. 3). (iii) The structure of an apo HTNV-L WT hexamer, determined at a resolution ranging from 3 to 9 Å (overall resolution 3.6 Å), reveals a peculiar configuration of polymerases into a trimer of dimers with a twofold C2 symmetry that had never been reported for any RNA-dependent RNA polymerase described so far (Fig. 1e and Supplementary Fig. 4). The central dimer acts as an anchor on which two external dimers bind symmetrically. While the central dimer displays a unique configuration, the two external dimers adopt the same conformation as the isolated dimer.

### Domain organization of HTNV-L

The HTNV-L polypeptide chain begins with the N-terminal ENDO domain and includes residues 1 to 220 (Fig. 2a, b). Its folding is relatively conserved compared to the isolated ENDO X-ray crystal structure that was comprising the residues 1 to 179[16], except for residues 1-19 that adopt a new conformation due to the presence of the residues 180 to 220 (rmsd 0.9 Å on 123 Cα residues, Supplementary Fig. 5).

Following the ENDO is the polymerase core (221-1601), which adopts a conformation equivalent to the previously obtained apo HTNV-L$_{D97A}$ structure[8] (rmsd between HTNV-L$_{WT}$ and HTNV-L$_{D97A}$

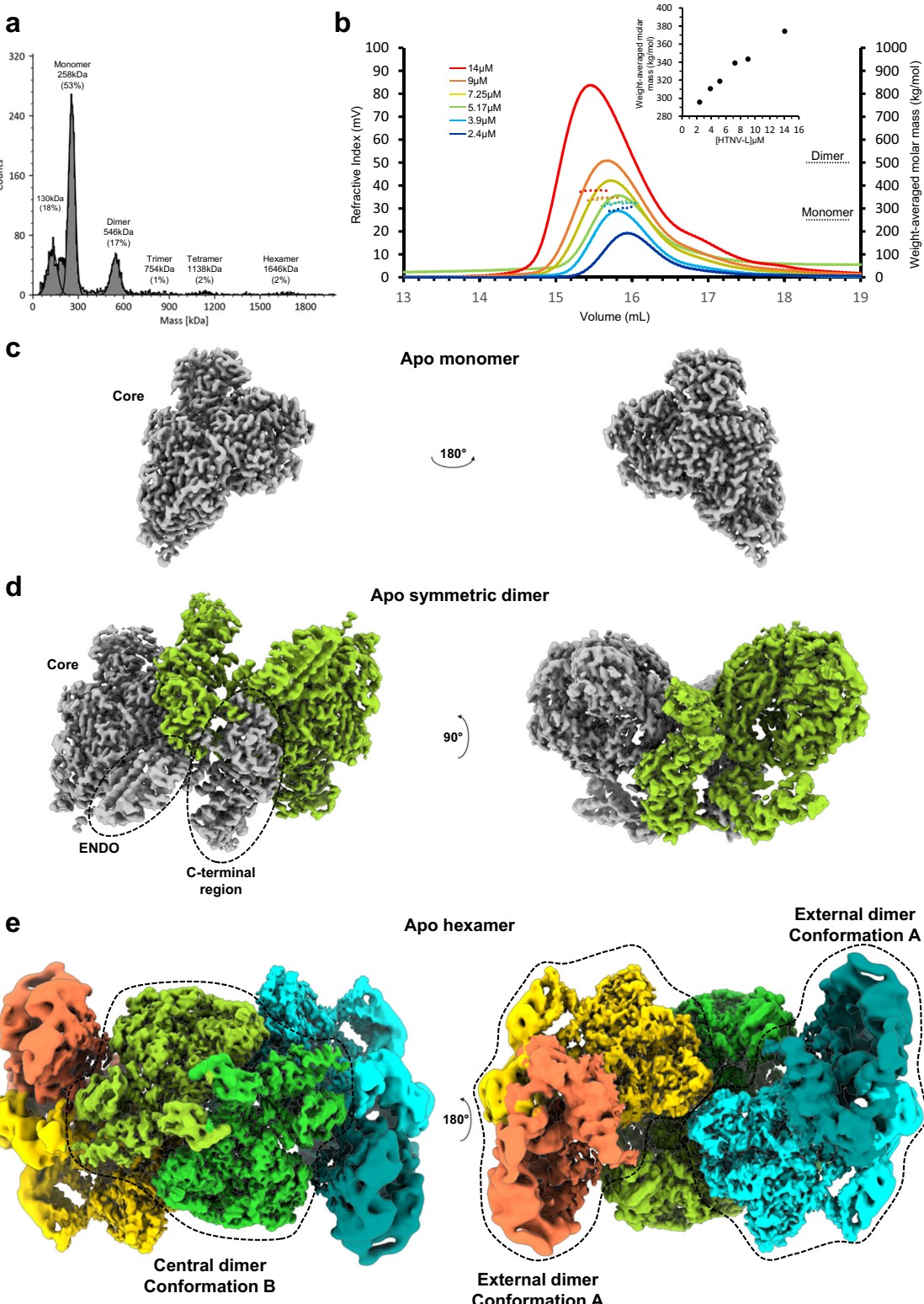

**Fig. 1 | Oligomeric states of HTNV-L. a** Mass photometry of HTNV-L in buffer containing 30 mM HEPES, 250 mM NaCl, and 5 mM TCEP. The molecular weight and the percentage of each species are indicated. **b** Size-exclusion chromatography and static light scattering (SEC-SLS) of HTNV-L injected at different concentrations, with one color attributed to each concentration. The elutions were monitored online using static light scattering and differential refractometry. The lines show the chromatograms monitored using differential refractive index measurements. The dotted lines indicate the molecular masses across the elution peaks calculated from static light scattering and refractive index. The numbers indicate the weight-averaged molar mass (kg/mol) of the observed equilibrium between different multimers. The inset shows the weight-average molar mass plotted as a function of concentration. Source data are provided as a Source Data file. **c–e** Composite cryo-EM maps of apo HTNV-L monomer (**c**), symmetric dimer (**d**), and hexamer (**e**). Each protomer is colored differently. Dimers are labeled according to their respective conformations, named A and B.

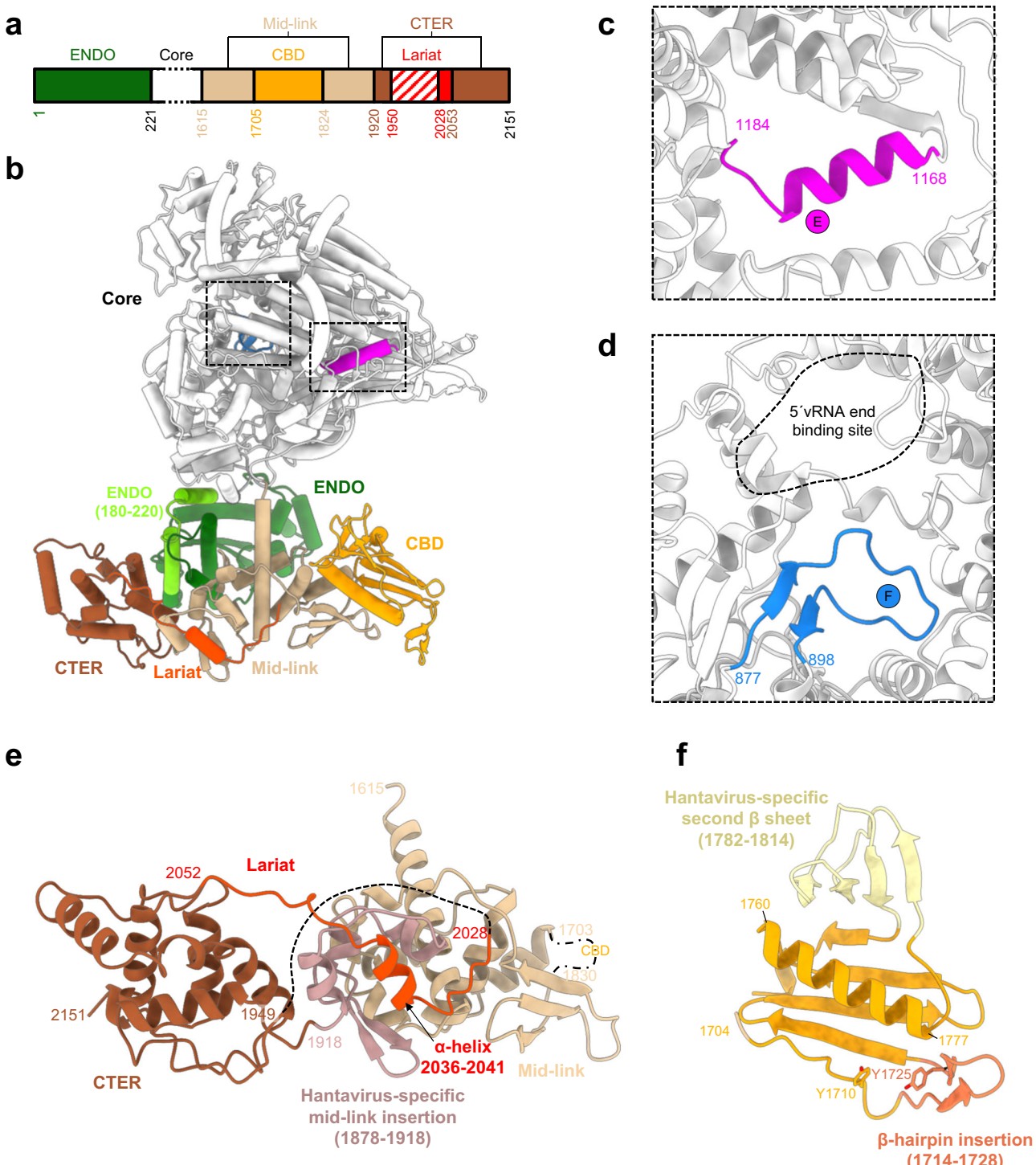

**Fig. 2 | Protomer organization of HTNV-L. a** Schematic representation of HTNV-L domain structure. **b** Cartoon representation of HTNV-L protomer with the endonuclease (ENDO), the core, the mid-link, the cap-binding domain (CBD), the lariat, and the domain C-terminal (CTER) respectively colored in forest green, white, beige, orange, red, and brown. The motifs E and F are respectively colored in magenta and blue. **c** Zoom on the motif E colored in magenta. **d** Zoom on the motif F colored in blue. The 5′vRNA end binding site is identified with a dotted line. **e** Zoom on the mid-link, the CTER, and the lariat colored as in **a**. The Hantavirus-specific mid-link insertion is indicated and colored in light pink. The CBD position is indicated with a dashed-dotted line. The dotted line that links the CTER to the lariat represents the residues of the lariat that are not visible due to flexibility. **f** Zoom on HTNV-L CBD model predicted by Alphafold and validated by an 8 Å cryo-EM map. The central five-stranded β-sheet and the α-helix that are common to all sNSV CBD is shown in the orange cartoon. The β-hairpin insertion that is likely to be essential for cap binding is shown in dark orange. The tyrosines that are predicted to bind the cap are shown as sticks. The Hantavirus-specific second β-sheet is shown in light yellow.

monomer: 1.0 Å on 1275 Cα residues). The overall resolution improvement, from 3.1 Å HTNV-L$_{D97A}$ to 2.6 Å in HTNV-L$_{WT}$, results in a more accurate definition of several regions, notably a β-hairpin of the core-lobe region (residues 676–695) that, by analogy with other

Bunyavirus polymerases, could be involved in the stabilization of the C-terminal region (Supplementary Fig. 6). The comparison of all HTNV-L apo protomer cores reveal their structural similarity, independently of their oligomeric states. Their catalytic motifs display an inactive

configuration, with a motif E organized in an α-helix, as previously reported for monomeric apo HTNV-L$_{D97A}$ structure[8] (Fig. 2c). The motif F is rather flexible and positions itself in the template entry tunnel, thereby closing it and confirming the inactive configuration of HTNV-L apo (Fig. 2d and Supplementary Fig. 7). Few local rearrangements can, however, be observed between the oligomeric states, notably for the putative priming loops. These loops are named according to their role in replication initiation in Influenza polymerases[21], but are located away from the active site in HTNV-L, where they rather correspond to template exit plugs[8]. Their exact position is conserved in HTNV-L monomer and HTNV-L hexamer, but differs in HTNV-L dimer. Their conformational change is associated with the reorganization of the thumb-ring residues 1455–1463 from an α-helix in HTNV-L monomer/hexamer to a loop in HTNV-L isolated dimer (Supplementary Fig. 8).

After the polymerase core lies the C-terminal region that is divided in three structural domains: the mid-link (residues 1615–1704 and 1824–1919), the CBD (1705–1823), and the CTER (residues 1920–1949, 2053–2151) (Fig. 2a). From the CTER domain protrudes a long loop that will be called lariat region, in reference to the name given of a similar protrusion in DBV-L[22] (residues 1950–2052, ordered part 2028 to 2052) (Fig. 2a, e).

The mid-link contains an α-helical mid-region and a three-stranded β-sheet link. While the mid-region has a size comparable to an equivalent region in other Bunyavirus polymerases[10,12,22,23], the HTNV-L link encloses a hantavirus-specific insertion containing three additional small α-helices and a β-hairpin (Fig. 2e and Supplementary Fig. 9).

Inserted in the mid-link lies the CBD. As it does not interact with any other domain in HTNV-L dimers and hexamers, the CBD remains highly flexible, precluding its structural characterization at high resolution. Nonetheless, the 8 Å density enables an unambiguous fit of the CBD model that was predicted with high confidence by Alphafold (per-residue confidence score−pLDDT−comprised between 67 and 90, with an average score of 80, on a scale from 0 to 100) (Fig. 2f and Supplementary fig. 9). The predicted CBD model consists in a central five-stranded β-sheet containing a small β-hairpin insertion (1714–1728) between the first and the second strand. A long α-helix (residues 1760–1777) packs against the central β-sheet. In addition to this minimal fold that is found in all sNSV CBD structures determined so far[10,12,22,24–26], a second four-stranded β-sheet is predicted that is specific to Hantaviruses (Fig. 2f and Supplementary Fig. 9). The capability of sNSV CBD to bind capped RNA has been related to the length of the β-hairpin insertion[26]. HTNV-L CBD predicted model displays a length comparable with the one of LACV-L CBD and DBV-L CBD[10,22,23], suggesting that HTNV-L may be capable of cap-binding in certain conditions. The resolution of the CBD structure is not sufficient to identify the residues involved in cap binding; however, the comparison of the HTNV CBD model with the structures of other sNSVs CBD suggests that the residues Y1710 and Y1725 may stack the cap m$^7$GTP moiety (Fig. 2f and Supplementary Fig. 9). In the low-resolution CBD density of HTNV-L dimer, the β-hairpin insertion however appears to be disordered (Supplementary Fig. 10), suggesting that large conformational change of the C-terminal regions are likely to be necessary for β-hairpin stabilization and subsequent cap-binding, as observed for LACV-L[10,11].

Finally, the CTER domain is composed of 6 α-helices that intertwine and from which protrudes a long and extended loop called the lariat (residues 1950–2052, ordered part 2028 to 2052) (Fig. 2e). The lariat α-helix 2036 to 2041 interacts with the α-helical part of the hantavirus-specific insertion of the mid-link, strengthening the mid-link/CTER interaction.

## HTNV-L protomers can adopt two drastically different conformations

When analysing the protomers that compose the apo symmetric dimer and the hexamer, it is striking to observe that they adopt two very different configurations (Supplementary Movie 1). The protomers that compose the isolated symmetric dimer and the external protomers of the hexamer globally adopt the same conformation, named conformation A, whereas the protomers of the central dimer of the hexamer adopt another configuration, named conformation B (Figs. 1d, 3). Superimposing the cores of conformers A and B reveals that the orientation of their respective ENDO differs by 68° (Supplementary Movie 1). Their entire C-terminal region also differs in position, with the mid-link domain that acts as a pivot point and modifies its orientation by 89° between conformers A and B (Fig. 3a and Supplementary Movie 1). Within the C-terminal region, the relative position of the domains also differs between the two protomer conformations: superimposing the mid-link of protomers A and B reveals that their CTER differ both in orientation (79°) and position (50 Å apart), while the CBD position is shifted by around 25 Å (Fig. 3b). As a result, the ENDO and the C-terminal region of protomers A and B exhibit very different mode of interaction with the core domain. The ENDO of both conformers interact with the lid domain, although this interaction involves different residues (Supplementary Fig. 11). The C-terminal region of conformer A is located away from its polymerase core, whereas the core and the C-terminal regions of conformer B tightly packs with its polymerase core, with the mid-link binding to the lid domain and the CTER interacting with the thumb and the thumb-ring domains (Fig. 3b and Supplementary Fig. 11).

## Presence of two different conformations of the protomers results in the formation of two different symmetric dimers

The divergence in the configurations of conformers A and B is compatible with the formation of two very different stable dimers.

The association of two protomers in conformation A results in the formation of symmetric dimer A that can be found either isolated or at the periphery of the hexamer (Fig. 1d, e). The solvent-accessible surface of one protomer buried at the dimer's interface, as estimated by PISA[27], is about 4200 Å$^2$, indicating a stable and intricate interaction involving numerous binding regions. The central hubs that promote symmetric dimer A formation are the CTER domains (Fig. 4a, b and Table 1). Of crucial importance is the extreme HTNV-L C-terminal loop containing the residues 2142 to 2151 that swaps into a hydrophobic groove of the opposite CTER (Fig. 4b), forming specific interactions that notably involve the two last HTNV-L residues F2150 and Y2151. F2150 is located in a hydrophobic cleft of the facing protomers where it stacks with P2120 and Y2064 (Fig. 4b). The aromatic ring of the C-terminal residue Y2151 forms hydrophobic interactions with the aliphatic moieties of N2110 and R2063, and the hydroxyl group of Y2151 makes hydrogen bonds with R2063 and D2106 (Fig. 4b). In addition, the interaction between CTER domains is reinforced by the binding of R2139 from protomer 1 with C2119 and D2121 of protomer 2 (Fig. 4b, c). Importantly, most of the residues involved in the interactions between the CTER domains are conserved amongst Hantaviruses (Supplementary Fig. 12 and Supplementary Data 1), suggesting that this interaction is likely to be biologically relevant and maintained during evolution.

The CTER domains are also involved in stabilizing the different domains that compose the dimer A. The CTER residues 2052–2056 and 2111–2127 from one protomer interact with the ENDO residues 185–204 of the other protomer, leading to a mutual stabilization of these regions (Fig. 4c and Table 1). In addition, the lariat residues 2044–2052, together with the mid-link 1905–1907 residues, make hydrophobic interactions with the core residues 1357–1364 and 1513–1514 of the other protomer (Fig. 4d and Table 1).

The only interaction between the two HTNV-L monomers that do not involve the CTER are mediated by few residues of the core. A symmetric interaction is notably observed between L1294 from one monomer with S1297 from the other, while S1444 and Q1445 interact with Y1543 and R1544 (Fig. 4e and Table 1). The buried interface remains minimal (16%) and the residues that interact are not conserved

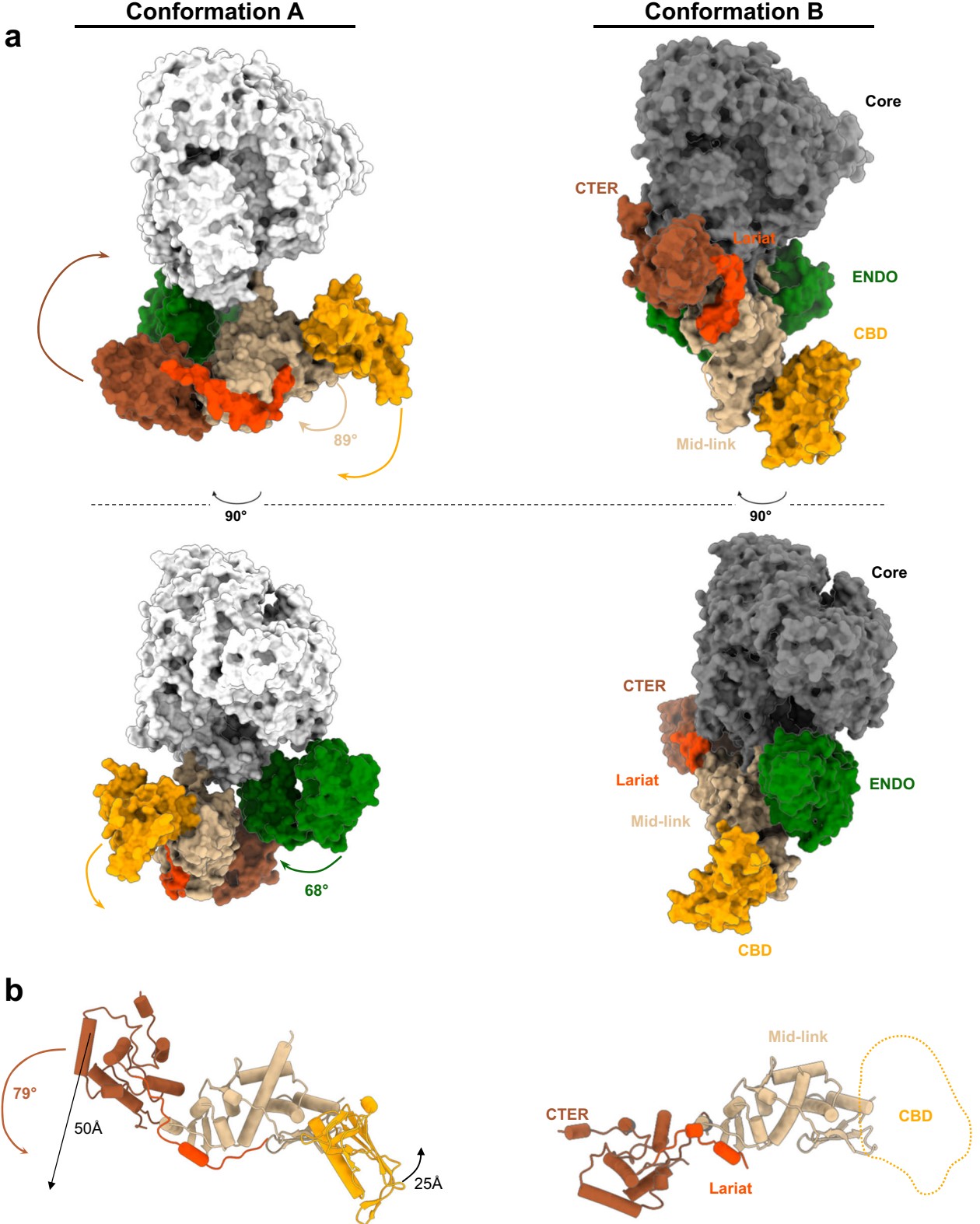

**Fig. 3 | Comparison of the two different conformations of HTNV-L protomers.** **a** Surface structure colored as in Fig. 2 showing the conformation of protomer A, present in the isolated apo dimer and in the external dimers of the hexamer, and protomer B, present in the central dimer of the hexamer. The rotations between both conformations are indicated. Two views that differ by 90° are displayed. **b** Zoom on the conformation of the C-terminal region of conformers A and B. Domain rotations and translations are indicated. A dotted line represents the position of the cap-binding domain (CBD) of the central dimer.

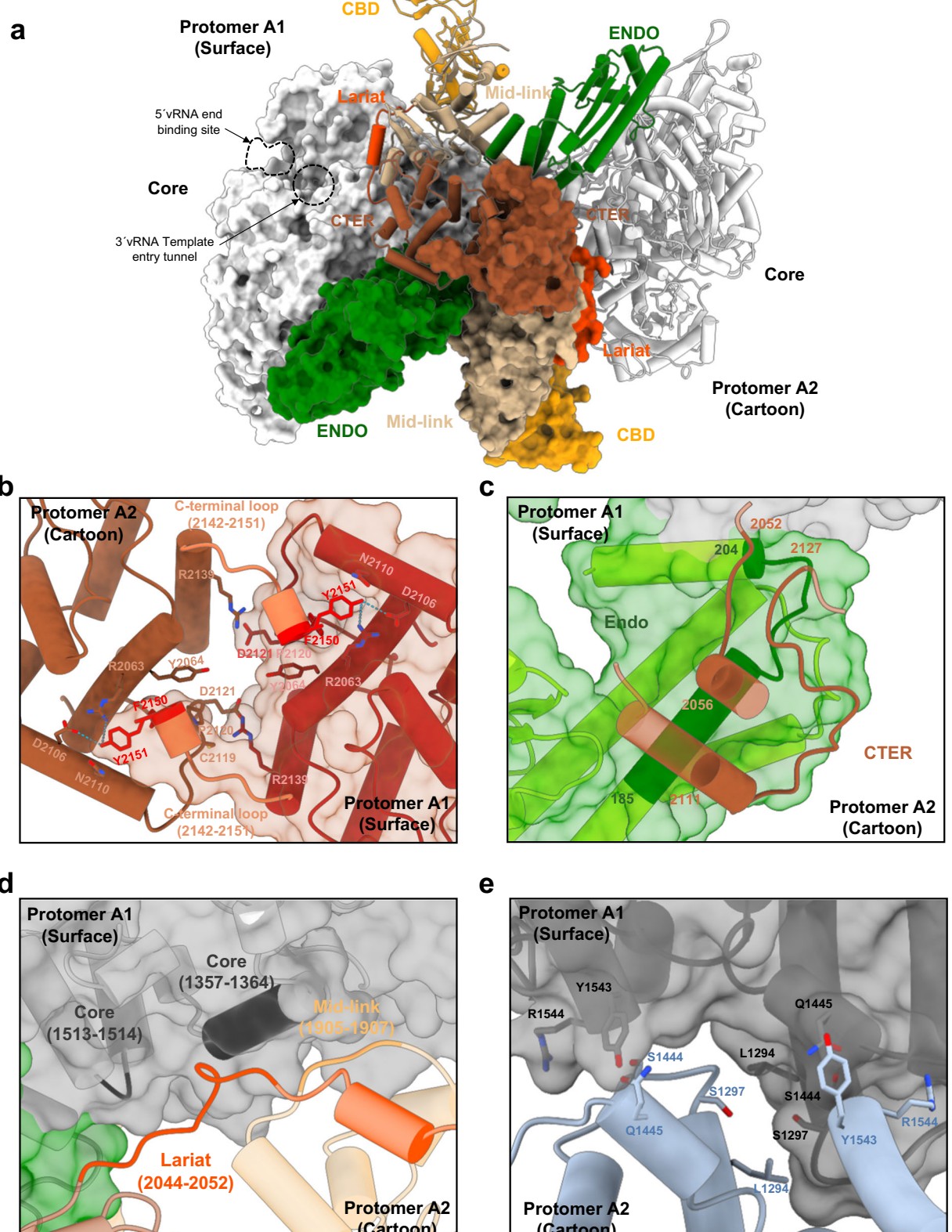

amongst Hantaviruses, suggesting the limited impact of CORE-CORE interaction in dimer formation.

This intricate organization is stabilizing most of the domains, except few regions that are located outwards of the dimer, such as the CBD and the external part of the ENDO. These domains display a higher degree of mobility and were not clearly discernible in the global 3D reconstruction map. Hence, image processing involving signal

subtraction, masking, and local refinement were necessary to visualize their organization (Supplementary Fig. 3).

The symmetric dimer A can be compared with the symmetric dimer generated by the interaction of protomers in conformation B. This second type of stable dimer constitutes the central dimer of the hexamer (Fig. 1d). In dimer B, each protomer buries about 2700 Å² of solvent-accessible surface at the dimer's interface, which is less than

**Fig. 4 | Organization and protomer interactions of HTNV-L apo symmetric dimer. a** HTNV-L apo symmetric dimer structure with protomer A1 shown as surface and protomer A2 shown as a cartoon. The domains are colored as in Fig. 2. The 5′vRNA end binding site and the template entry tunnel, to which the 3′vRNA binds, are indicated. **b** Zoom on the interaction of the C-terminal domains (CTER) in the same orientation as in (**a**). Protomer A1 surface is shown in transparent dark red, and its cartoon representation is in dark red. Protomer A2 is shown as a brown cartoon. Residues in interaction are shown as sticks. The C-terminal loops 2142–2151 that are crucial in the C-terminal swapping stabilization are shown in coral, with F2150 and Y2151 shown in red. **c** Zoom on the interaction of the endonuclease (ENDO) of protomer A1 with the CTER of protomer A2. Protomer

A1 surface and cartoon are colored as in (**a**) with the regions that interact shown in dark green with residue numbers indicated. The CTER regions of protomer A2 that interact with protomer A1 are displayed as brown cartoons, the regions that interact are indicated. **d** Interaction of the lariat and the mid-link of protomer A2 with the core of protomer A1. Both protomers are colored as in (**a**) and shown as semi-transparent. The regions of the lariat, the mid-link, and the core that interact are respectively shown in dark orange, orange, and dark gray. Residue numbers of the regions that interact are indicated. **e** Interactions of the core of protomer A1, shown in gray cartoon and surface, with the core of protomer A2, shown as a light blue cartoon. The interacting residues are shown as sticks and are numbered.

the one in dimer A, but still significant (Table 1 and Fig. 5a). Despite the large differences in the arrangement of the protomers present in dimers A and B, it is interesting to note the presence of some similarities in the regions that contribute the most to the interface. In both cases, the CTER domains remain the main hub of the protomer interactions (Fig. 5a and Table 1), that notably involve the swapping of C-terminal residues 2142–2151 into the CTER hydrophobic groove of the second protomer (Figs. 4b, 5a). However, some protomer-protomer interactions are specific to dimer B. For instance, the CTER α-helix residues 2131–2141 and the CTER loop 2118–2121 of one protomer B interacts with their symmetrical equivalent. Moreover, the CTER being in contact with the thumb-ring and the thumb is bolstered through interactions with the core of the second protomer B (Fig. 5a and Table 1). In addition, the polymerase cores of the two protomers B make hydrophobic contacts through their respective thumb domains, although their contribution to the dimer's interface remains modest (19%) (Fig. 5a and Table 1). Finally, the lariat residues interact with the polymerase core of the other protomer (Table 1).

### Table 1 | Analysis of the interaction surfaces within apo HTNV-L protomers

| a Dimer A | | | |
|---|---|---|---|
| **Protomer A1** | **Protomer A2** | **Surface of interaction (Å²)** | **Percentage** |
| CTER A1 | CTER A2 | 1093 | 26.1% |
| CTER A1 | ENDO A2 | 794 | 18.9% |
| ENDO A1 | CTER A2 | 794 | 18.9% |
| Core A1 | Core A2 | 671 | 16.0% |
| Core A1 | Lariat A2 | 234 | 5.6% |
| Lariat A1 | Core A2 | 234 | 5.6% |
| Core A1 | Mid-link A2 | 185 | 4.4% |
| Mid-link A1 | Core A2 | 185 | 4.4% |
| | Total: | 4190 | 100% |
| **b Dimer B** | | | |
| **Protomer B1** | **Protomer B2** | **Surface of interaction (Å²)** | **Percentage** |
| CTER B1 | CTER B2 | 1009 | 37.8% |
| Core B1 | Core B2 | 507 | 19.0% |
| Core B1 | CTER B2 | 320 | 12.0% |
| CTER B1 | Core B2 | 320 | 12.0% |
| Core B1 | Lariat B2 | 199 | 7.4% |
| Lariat B1 | Core B2 | 199 | 7.4% |
| Core B1 | Mid-link B2 | 59 | 2.2% |
| Mid-link B1 | Core B2 | 59 | 2.2% |
| | Total: | 2672 | 100% |

**a, b** The domains that interact within protomers of dimer A (**a**) and protomers of dimer B (**b**) are indicated along with their interacting surfaces. Percentages quantify each specific interaction compared to the global interaction between protomers.

### Interactions between HTNV-L dimers A and B result in the formation of HTNV-L hexamer

Dimer B is observed only when capped by two dimers of type A (Fig. 5b). Binding of dimers A to dimer B is directional, the same side of both external dimers A being in contact with the central dimer B, and induces minimal reorganization of dimers A in comparison with their conformation in the isolated dimer form (Supplementary Fig. 13).

The symmetric hexamer formed cannot be further elongated into larger complexes using the known protomer conformation and the same dimer A – dimer B interface. One can, however, imagine that some sub-complexes could exist, that would contain less than six subunits. This would be consistent with the observation of trimers and tetramers visualized by mass photometry (Fig. 1a).

For complete hexamers, the interaction between protomers is extensive, reaching about 15,600 Å² (Fig. 5b and Table 2). Analysis of the interface area between protomers confirms that the hexamer is indeed composed of a trimer of dimers as dimer B and both dimers A display large protomer interfaces, respectively around 3900 Å² and 2500 Å², whereas the other protomer-protomer interfaces found in the hexamer are smaller, comprised between 350 to 1351 Å² (Table 2). However, the interface between the two types of dimers reaches 2300 Å², supporting the stability of the hexameric form.

### Incubation with 5′viral RNA disrupts HTNV-L apo multimers and results in the formation of another type of HTNV-L dimer

To analyze if incubation with 5′vRNA is modifying the equilibrium between the different oligomers, mass photometry was performed in buffers containing different ionic concentrations (Fig. 6a and Supplementary Fig. 14). The results are very similar to the ones obtained in the absence of 5′vRNA except for the hexamers that could not be automatically detected. The 150 kDa species is also less abundant, suggesting that the 5′vRNA stabilizes HTNV-L, partially preventing its degradation.

SEC-SLS of HTNV-L incubated with 5′vRNA results in a single peak with a weight-averaged molar mass that varies from 269 to 382 kDa when HTNV-L concentration is increased from 1.36 to 6.37 μM (Fig. 6b).

To analyze if incubation with 5′vRNA is modifying HTNV-L conformation and oligomerization mode, a cryo-EM dataset was collected in the presence of 5′vRNA. Intriguingly, cryo-EM 2D class averages reveal that only 0.3% of symmetric dimers and 0.2% of symmetric hexamers are detected in the presence of 5′vRNA, the extremely low number of particles preventing further structural characterization (Fig. 6c). This strongly differs from the 2D class averages visualized in the absence of vRNA in which 10.1% and 2.1% of the particles correspond to symmetric dimers and hexamers (Fig. 6c). We could speculate that the symmetric multimers detected in the dataset collected with 5′vRNA could correspond to the few particles that would have remained in their apo form. Interestingly, another type of dimer is detected in 2D class averages of the HTNV-L dataset collected in the presence of 5′vRNA (Fig. 6c). We will call this dimer "5′vRNA-related HTNV-L dimer". When one polymerase core from the apo symmetric

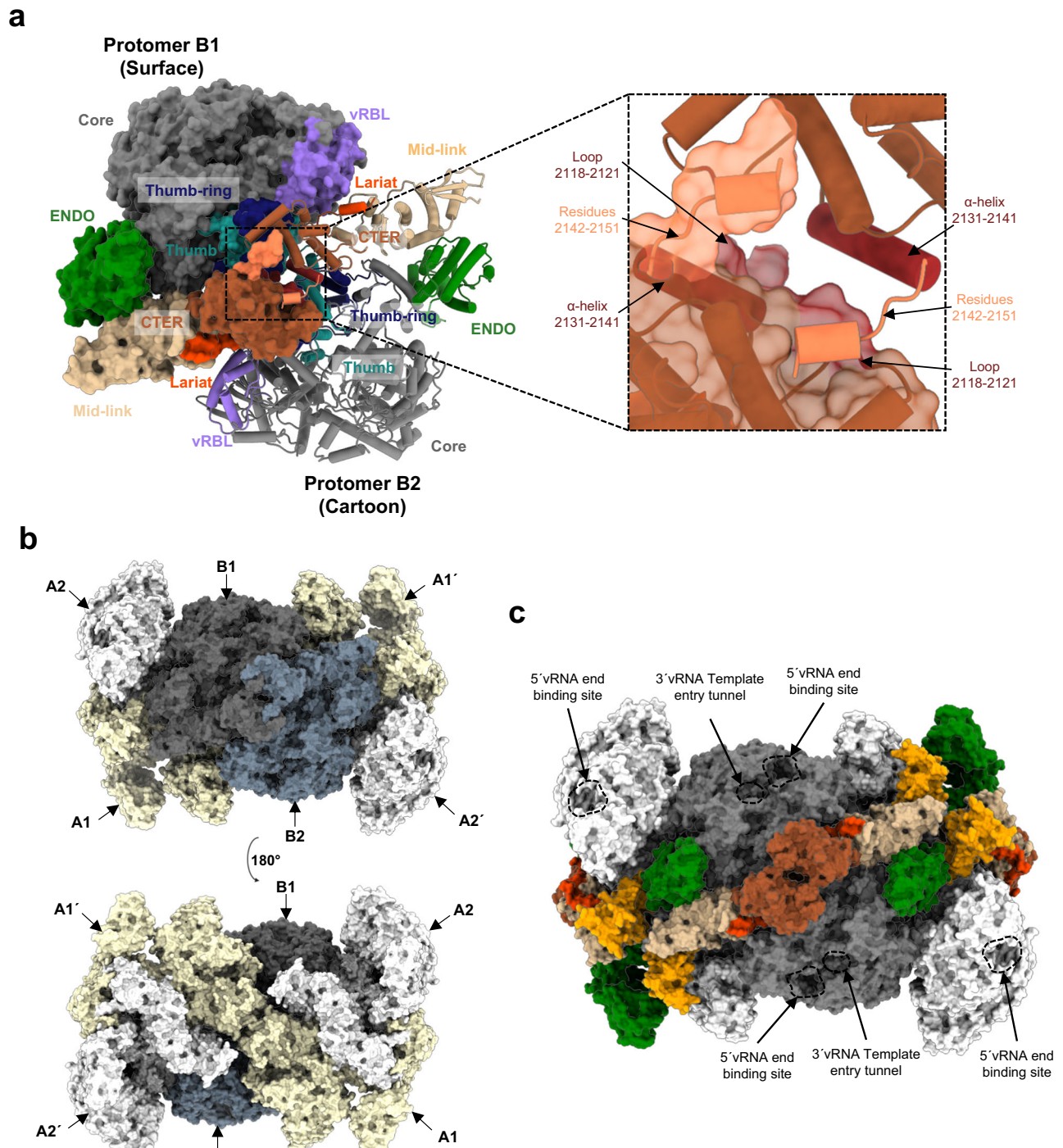

**Fig. 5 | HTNV-L apo hexameric structure: interactions between protomers.**
**a** Interaction of two protomers B results in the formation of dimer B that corresponds to the central dimer of the hexamer. One protomer is shown as a surface and the second as a cartoon. The ENDO, the mid-link, the CTER, and the lariat are colored as in Fig. 2. The vRBL, the thumb, and the thumb-ring are respectively colored in purple, light sea green and midnight blue. On the right side, zoom on the C-terminal domain (CTER). Interacting residues of the CTER are indicated. Residues 2142–2151 that swap into the other protomer are displayed in orange. **b** Surface

representation of the hexamer visualized in the same orientation as in a (top) and rotated by 180° (bottom). The protomers B1 and B2 are colored as in (**a**), the external protomers A1 and A1' are colored in white, the external protomers A2 and A2' in beige. **c** Surface of the hexamer structure with domains colored as in Fig. 2. The cores of the protomers of the central dimer are colored in gray, while the cores of the protomers of the external dimers are colored in white. The 5'vRNA end binding site and the template entry tunnel, to which the 3'vRNA bind, are indicated.

dimer is shown in the same orientation as one polymerase core of the 5'vRNA-related HTNV-L dimer, it clearly appears that the orientation of their second core differs (Fig. 6d). This, combined with the visualization of extra densities that may correspond to the ENDO and/or the C-terminal regions, suggests that the 5'vRNA-related HTNV-L dimer

could be asymmetric (Fig. 6e). The dimers detected in the presence of 5'vRNA represent 7.6% of the particles, a percentage roughly comparable to the 10.1% of symmetric dimers visualized in the absence of vRNA, explaining why these two different types of dimers could not be distinguished by mass photometry and SEC-SLS due to their similar

**Table 2 | List of the interactions between protomers in the context of the hexamer**

| Protomer in interaction | Surface of interaction (Å²) | Domain involved in interaction | |
|---|---|---|---|
| A1 - A2 = A1' - A2' | 3936 | ENDO / Thumb / Thumb-ring / Lid / Mid-link / Lariat / CTER | |
| B1 - B2 | 2539 | vRBL / Thumb / Thumb-ring / Lariat / CTER | |
| B1 - A1' = B2 - A1 | 1351 | B1 (or B2): Core-lobe / vRBL | A1' (or A1): Core-lobe |
| B1 - A1 = B2 - A1' | 639 | B1 (or B2): ENDO / linker / Fingers | A1 (or A1'): Thumb / Thumb-ring / Core-lobe |
| A1 - A1' | 558 | vRBL / Arch | |
| B1 - A2 = B2 - A2' | 351 | B1 (or B2): ENDO | A2 (or A2'): Thumb / Thumb-ring / CBD |
| | Tot: 15651 | | |

Protomers are named according to Fig.5b. Their interaction surface is indicated, as well as the domains involved.

masses. Unfortunately, the 5′vRNA-related HTNV-L dimers adopt a strong preferential orientation on the cryo-EM grid, preventing their 3D characterization. These preliminary results will need to be confirmed by future work that should solve the strong preferential orientation problem and determine the high-resolution 3D structure of the 5′vRNA-related HTNV-L dimers.

### Conformational changes of HTNV-L monomeric structures induced by 5′vRNA-binding

To visualize the potential conformational changes that could explain the disruption of symmetric oligomers and the formation of the 5′ vRNA-related dimer, we further analyzed the HTNV-L dataset collected in the presence of 5′vRNA. 5′vRNA-bound monomeric HTNV-L structures were determined at an overall resolution of 2.8 Å (Supplementary Fig. 15 and Supplementary Table 2). 3D classification revealed the presence of two distinct conformations of HTNV-L monomers that will be called "intermediate" and "active" (Fig. 7). In the intermediate conformation, 5′vRNA-binding induces local reconfiguration, notably in the motif F and in the linker connecting both the core-lobe and the finger domains. In addition, whereas very few global movements of HTNV-L are visible (Supplementary Fig. 16), 5′vRNA-binding induces several movements of the canonical RdRp motifs. The motif E retains its amphipathic α-helix configuration but changes its orientation compared to its position in apo HTNV-L (Fig. 7b). This alteration modifies the van der Walls interactions between motif E and the α-helix 929–948, inducing a global movement of the α-helix 929–948 and the ordering of the putative PR loop. Minor movements are also observed in motifs A, C, and D, although no magnesium ion is visible in the active site.

The reconfiguration of 5′vRNA-bound HTNV-L into a conformation compatible with activity induces a more drastic global opening of the polymerase, which is associated with the large reorganization of the motif E from an amphipathic α-helix to a three-stranded β-sheet (Supplementary Fig. 16). Whereas movements within motif E are significant, with large movements of each of its amino acid, the movements of the other motifs remain relatively small compared to the intermediate conformation (Fig. 7). The reorganization of motif E induces the formation of a pocket and the binding of a magnesium ion that is coordinated between D972 of motif A, D1099 of motif C and E1170 of motif E (Fig. 7).

When superimposing the 5′vRNA-bound monomeric HTNV-L structure onto the HTNV-L apo dimeric structure (Supplementary Fig. 17), we observe that the global opening of HTNV-L core induced by 5′vRNA-binding modifies the orientation of the thumb, the thumb-ring, the lid and the bridge domains that would result, in apo symmetric dimers, in clashes between the bridge domain of one protomer and the lariat region of the second. These conformational changes could explain the displacement of the equilibrium visible in the apo form (monomer, symmetric dimer, and symmetric hexamer) towards a new equilibrium in the presence of 5′vRNA (monomer and a yet-to-be characterized 5′vRNA-related dimer) (Fig. 8).

## Discussion

### Comparison of HTNV-L C-terminal region with other *Bunyavirales* polymerases

The structures of apo HTNV-L dimer and hexamer reveal the organization of the C-terminal region of *Hantaviridae* polymerase (Figs. 1, 2). It is striking to observe that, although the C-terminal region is quite divergent in terms of amino acid sequence amongst Bunyavirus polymerases, the C-terminal domains organization of all *Bunyavirales* polymerases described so far is rather conserved (Supplementary Fig. 18). Bunyavirus polymerases all contain a mid-link domain, that is positioned centrally and links the polymerase core to the C-terminal region. The CBD is inserted in the mid-link and is also present in all *Bunyavirales* polymerases determined so far. The CTER domains display the largest variability among bunyaviruses, but, interestingly, they all contain a large protrusion that is named β-hairpin strut in LACV-L, and lariat in BDV-L and HTNV-L. In LACV-L and BDV-L, this protrusion stabilizes the relative position of the C-terminal region, and the core by interacting with a specific β-hairpin present in the core-lobe[10,22] (Supplementary Fig. 6). In HTNV-L, the core-lobe β-hairpin is conserved, and one can hypothesize that the region of the lariat that is not visible in the HTNV-L current structures might bind to the core-lobe β-hairpin and stabilize the entire HTNV-L structure (Supplementary Fig. 6). In HTNV-L, the visible part of the lariat, in addition, stabilizes the mid-link and therefore contributes to the steadying of the entire C-terminal region.

Altogether, these observations clearly indicate that *Bunyavirales* polymerases have conserved a common organization during evolution, even in their C-terminal region that diverges the most in terms of sequence.

### Comparison of *Bunyavirales* polymerase dimers: diversity in structure and in function

The HTNV-L apo structures reveal the formation of two very different symmetric dimers that display large interface areas. Interestingly, both dimers involve symmetric interactions of their CTER domains, with the swapping of the C-terminal residues of one protomer into the second. The conservation of this binding mode suggests the importance of the CTER domains in polymerase dimerization. Interestingly, the involvement of the CTER domains and the C-terminal α-helix swap had already been observed in LACV-L dimers (Supplementary Fig. 19a)[10]. In this previous study, the physiological relevance of LACV-L dimers remained to be ascertained as, although they had been observed in the asymmetric unit of the C2 crystal form, the crystal packing may had induced significant conformational changes. The HTNV-L observations thus reinforce the hypothesis that the involvement of the CTER domain in symmetric dimer formation is physiologic and might be conserved amongst Bunyavirus polymerases.

In contrast, a comparison of HTNV-L and MACV-L symmetric dimers[12,14,15] clearly indicates the difference in dimer configuration, although protomer interactions involved in both cases, the ENDO and the thumb domain of the polymerase core (Supplementary Fig. 19b).

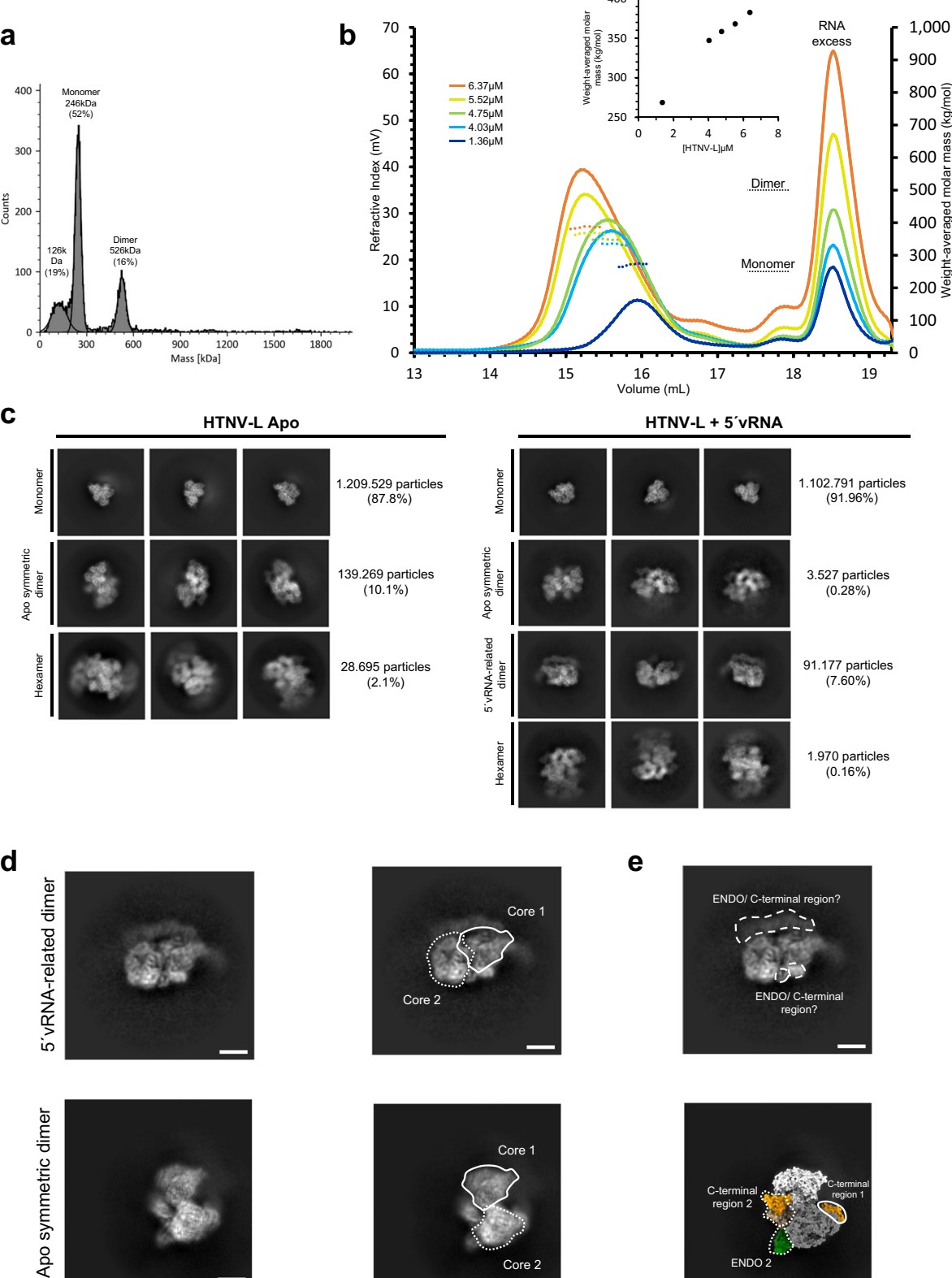

Interestingly, 3'vRNA-binding in MACV-L results in the stabilization of the symmetric dimers that become the major species, whereas 5'vRNA-binding in HTNV-L destabilizes the symmetric dimers. In addition, we here observe that incubation with 5'vRNA results in the formation of another type of 5'vRNA-related HTNV-L dimer. The identification of such a dimer is intriguing and opens the way to future investigations that will decipher its exact 3D structure, the conformational changes necessary for its formation, and its functional role in replication and transcription.

The diversity of polymerase multimers observed within the *Bunyavirales* order, and the modification of their configuration suggests that different types of dimers and multimers are likely to exist during the replication and transcription cycles, that would be related to different yet-to-be-determined biological roles.

**Fig. 6 | Incubation with 5′vRNA disrupts HTNV-L symmetric oligomers and results in the formation of another type of 5′vRNA-related HTNV-L dimers.** **a** Mass photometry of 5′vRNA:HTNV-L ratio 10:1 in buffer containing 30 mM HEPES, 250 mM NaCl, and 5 mM TCEP. The molecular weight and the percentage of each species are indicated. **b** Size-exclusion chromatography and static light scattering (SEC-SLS) of 5′vRNA:HTNV-L ratio 10:1 injected at different concentrations, with one color attributed to each concentration. The elutions were monitored online using static light scattering and differential refractometry. The dotted lines show the chromatograms monitored using differential refractive index measurements. The dotted lines indicate the molecular masses across the elution peaks calculated from static light scattering and refractive index. The numbers indicate the weight-averaged molar mass (kg/mol) of the observed equilibrium between different multimers. The inset shows the weight-average molar mass plotted as a function of concentration. Source data are provided as a Source Data file. **c** 2D class averages of

HTNV-L apo (left) and HTNV-L incubated with 5′vRNA (right). Monomer, symmetric dimer, 5′vRNA-related dimer, and hexamer class averages are shown. The number of particles and the percentage compared to the entire dataset is indicated. **d** 2D class averages of the 5′vRNA-related dimer (top) and the apo symmetric dimer (bottom). The core of protomer 1 is shown in the same orientation in both 2D classes and is surrounded by a white solid line (right). Protomers 2 adopt a different position in both dimers and the core of protomer 2 is surrounded by a short dotted line (right). **e** For the 5′vRNA-related dimer (top), the extra densities that may correspond to positions of the endonuclease (ENDO) and C-terminal regions are indicated with white elongated dotted lines. For the apo dimer (bottom), a projection of the structure in the 2D class orientation is shown with the cores colored in white and gray, the ENDO in green, and the C-terminal regions in beige and orange. The ENDO/C-terminal regions of protomer 1 are surrounded by a white solid line, equivalent regions of protomer 2 by short dotted lines.

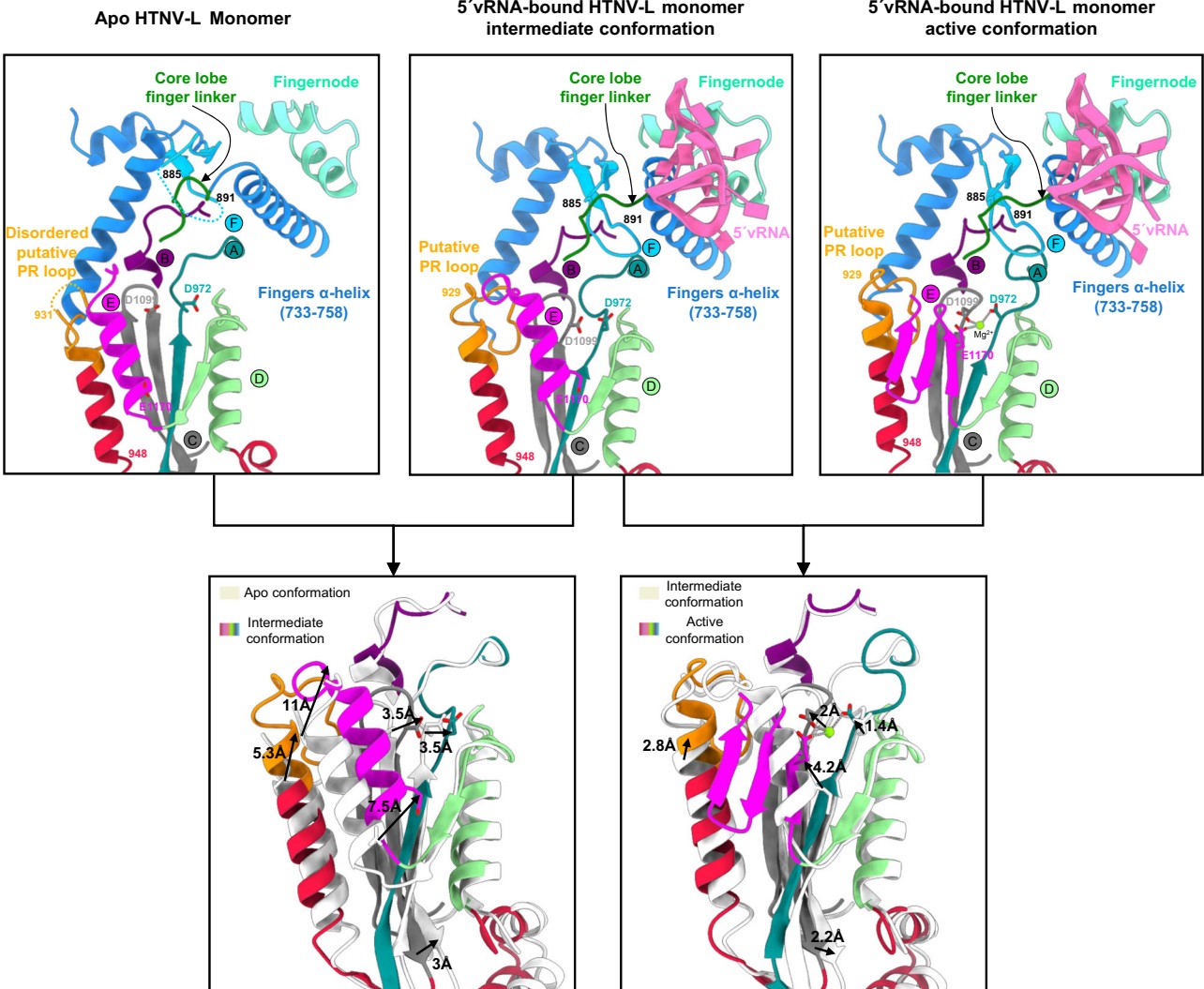

**Fig. 7 | Conformational changes in the HTNV-L active sites between apo, 5′ vRNA-bound intermediate, and 5′vRNA-bound active conformations.** Zoom on the active sites of monomeric apo HTNV-L on the top left, 5′vRNA-bound intermediate conformation in the top middle and 5′vRNA-bound active conformation on the top right. The motifs, the putative prime-and-realign loop (PR loop), the 5′

vRNA-binding site are labeled and have specific colors. Superimposition of monomeric apo HTNV-L with 5′vRNA-bound intermediate conformation (bottom left) and superimposition of 5′vRNA-bound intermediate conformation with 5′ vRNA-bound active conformation (bottom right) are shown to highlight motif movements.

## HTNV-L hexamers: a peculiarity never observed for polymerases

The formation of polymerase hexamers had never been reported for any sNSV polymerases. To our knowledge, the only other report describing high-resolution structures of large polymerase multimers

concern single-stranded positive sense Picornaviruses, such as Polio-virus or Foot-and-Mouth-disease virus, that replicate at the cytosolic surface of cytoplasmic membranes where they form planar and tubular oligomeric arrays[28,29]. These arrays correlate with cooperative RNA-

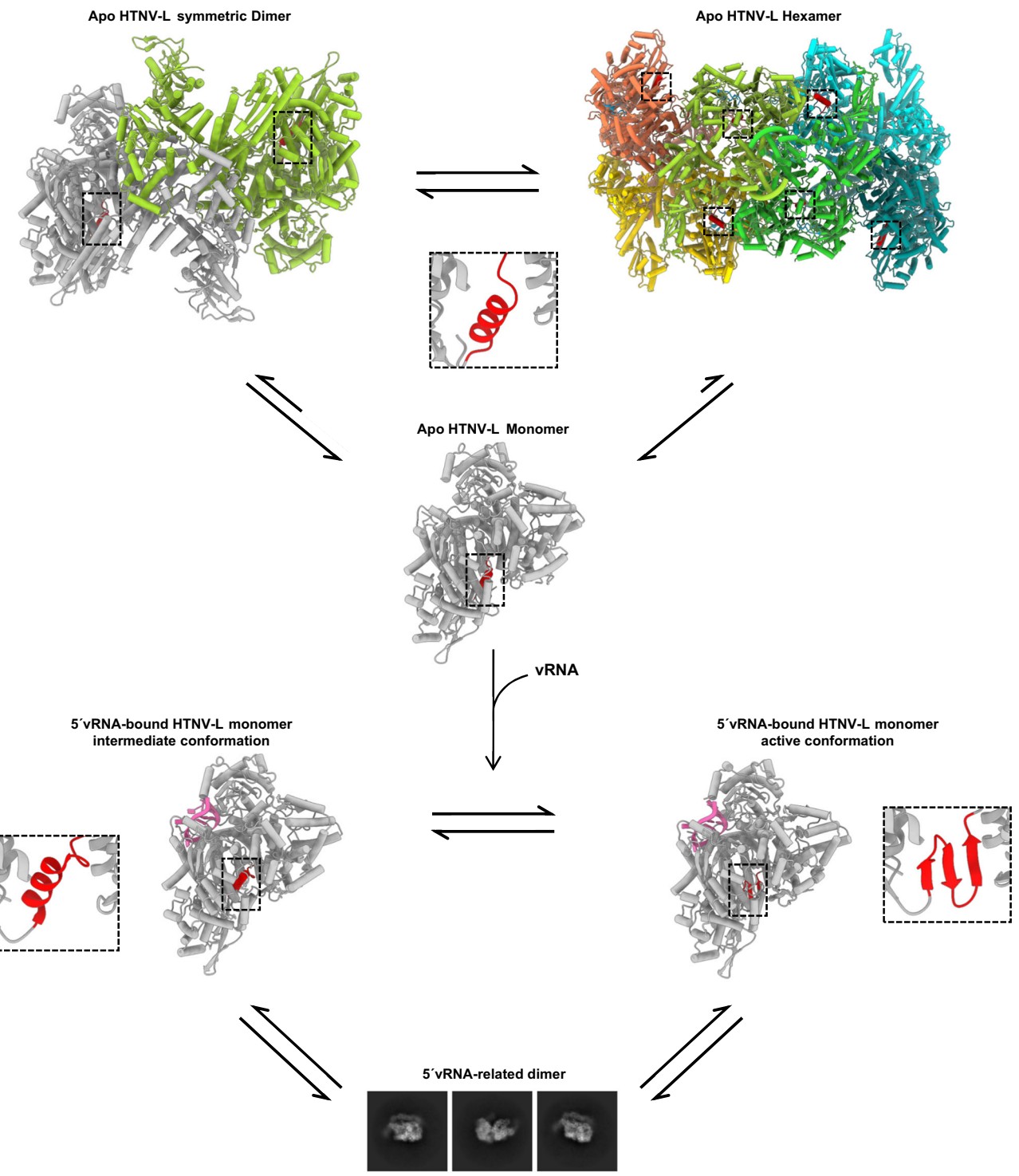

**Fig. 8 | Schematics of HTNV-L species in equilibrium in the absence and presence of 5′vRNA.** HTNV-L apo is in equilibrium between symmetric dimers, hexamers, and monomers, as shown by the cartoon representation of HTNV-L colored by protomers. Arrows indicate the equilibrium between the different oligomers. Motif E is shown in red and is surrounded by a dotted line. In each protomer of HTNV-L apo, motif E is in an α-helical conformation as shown by a zoom where motif E is colored in red. Incubation with 5′vRNA disrupts the symmetric oligomers binding and RNA elongation[30]. These multimers are thus in strong contrast with the observed HTNV-L hexamers that are visualized only in the absence of RNA. In addition, HTNV-L hexamers cannot be further elongated since the relative orientation of dimer A C2 axis and dimer B and results in an equilibrium between monomers and another type of 5′vRNA-related dimer. Two different conformations of 5′vRNA-bound monomeric HTNV-L co-exist that are displayed as a white cartoon with the 5′vRNA displayed in light pink. The conformation of motif E in both conformations is shown in red and is surrounded by a dotted line. A zoom on motif E shows its conformation in each map. The 3D structure of the 5′vRNA-related dimer remains to be characterized. 2D class averages of the 5′vRNA-related dimer are displayed.

C2 axis is not compatible with the binding of other dimers. It remains to be investigated if the observed hexameric polymerase is specific to Hantaviruses or if it also exists in other *Bunyavirales* in specific conditions.

## Hypothetical role of apo HTNV-L dimers and hexamers

In vitro apo symmetric dimer and hexamer assemblies occur in buffers that are typical to structural and functional assays, pH 8 in the presence of 250 mM NaCl, that are not far from physiological conditions. The visualization of the multimers in vitro opens the question of their presence and their function during cell infection. The field of research opened by our results is large and fascinating as Bunyavirus polymerases fulfill multiple functions during cell infection and, therefore, as described in the precedent paragraph, are likely to form several types of dimers and multimers that would have different specific roles during the viral cycle. It would be interesting to identify if the apo symmetric dimers and hexamers observed here can become the majority in certain conditions and which elements or conditions would stabilize them. One can think of specific compartmentalization that would locally increase the polymerase concentration and would thus favor oligomerization. Another possibility is that host-cell co-factors may be necessary to stabilize these multimers. If symmetric apo dimers and hexamers observed here become the major species at one specific state, one possible role that we speculate here is a storage and protection state of newly translated polymerases that have not yet been in contact with viral RNA. Apo dimer and hexamer formation would prevent the large movements of the ENDO and the C-terminal region observed in HTNV-L monomers, thereby stabilizing the entire polymerase structure. The accessibility of the 5′ and 3′ vRNA binding sites to each protomer, and the observed disruption of these multimers upon vRNA binding is compatible with this "polymerase storage" hypothesis.

If future research confirms the presence of these apo symmetric multimers during cell infection, our results would also provide a solid framework for the future development of antivirals. Indeed, molecules that would stabilize and enrich the inactive apo multimer population could be considered as possible leads that would preclude HTNV-L activation that is necessary for genome replication and transcription.

## Methods

### Cloning, expression, and purification

The full-length HTNV-L gene (strain 76-118/Korean hemorrhagic fever, GenBank: X55901.1 UniProt: P23456) flanked in 5′ by a sequence coding an N-terminal hexa-histidine tag was cloned in a pFastBac vector between NdeI and NotI restriction sites. The expressing baculovirus was prepared using the Bac-to-Bac method (Invitrogen)[31]. *Trichoplusia ni* (High Five™ Cells BTI-Tn-5B1-4 Invitrogen PN/51-4005 lot 1783124, none of the cell line used were authenticated) were infected at $0.7 \times 10^6$ cells/mL with 0.1% v/v of baculovirus and harvested 120 h after infection. The culture medium was centrifuged at $1000 \times g$ for 15 min. The cell pellets were resuspended in lysis buffer (30 mM HEPES pH 8, 300 mM NaCl, 10 mM Imidazole, 2 mM TCEP, and 5% glycerol) supplemented with cOmplete EDTA-free protease inhibitor complex (Roche) and ribonuclease A (Roche). The lysate was sonicated for 3 min 30 s (10 s ON, 20 s OFF, and 40% intensity) and centrifuged for 45 min at $20,000 \times g$ and 4 °C. The supernatant was filtered at 0.8 μm and used for purification by nickel ion affinity chromatography (GE Healthcare). A washing step in the lysis buffer supplemented with 30 mM imidazole was followed by the elution in lysis buffer supplemented with 500 mM imidazole. HTNV-L fractions were slowly diluted by two at 4 °C using the heparin buffer (30 mM HEPES pH 8, 300 mM NaCl, and 5 mM TCEP), loaded on a 1 ml heparin column (GE Healthcare), and eluted in the heparin buffer supplemented with 500 mM NaCl. For cryo-EM assays and some mass photometry assays, a final gel filtration step was performed in GF buffer (30 mM HEPES pH 8, 250 mM NaCl, 10 mM TCEP) using a S200 size-exclusion chromatography column (GE Healthcare).

### Mass photometry and size-exclusion chromatography coupled to static light scattering (SEC-SLS)

Mass photometry measurements were carried out on a OneMP mass photometer (Refeyn Ltd). Prior to sample preparation, the coverslip (No. 1.5H, 24 × 50 mm, VWR) was washed several times with water and isopropanol and dried with compressed air before being used as a support for the silicone gaskets (CultureWell™ Reusable Gaskets, Grace Bio-labs). For mass/contrast calibration, 19 μL of mass-photometry calibration buffer containing 30 mM HEPES pH 8, 250 mM NaCl, and 5 mM TCEP was deposited in a well of the silicone gasket and used to determine the focus. About 1 μL of native marker (Native Marker unstained protein standard, LC0725, Life Technologies) diluted 20x in the mass-photometry buffer was subsequently added to the 19 μL drop for mass/contrast calibration that was monitored during 60 s using the AcquireMP software (Refeyn Ltd, Version 2.3.0) with a medium field of view (detection area of 56 μm²). For HTNV-L measurements, 19.5 μL of the buffer corresponding to each experiment was deposited on a well of the silicone gasket and used to find the focus. About 0.5 μL of HTNV-L in the buffer corresponding to each experiment was added and gently mixed to reach a final concentration of 30 nM. For measurements in the presence of 5′vRNA, HTNV-L was mixed with 5′vRNA (5′- UAGUAGUAGACACCGCAAGAU-GUUA-3′) in a ratio 10:1 in a buffer corresponding to each experiment prior to mass photometry analysis. Data acquisition was carried out as for mass/contrast calibration. Data analysis was performed using DiscoverMP software (Refeyn Ltd, version 2.3.0).

SEC-SLS measurements were performed on an OMNISEC (Malvern). The SEC was performed with a superose 6 10/300 GL (GE Healthcare) equilibrated in GF buffer at a flow rate of 0.5 mL/min. The column was calibrated with a 50 μL injection of Bovine Serum Albumin at 2 mg/mL that identifies the monomer and the dimer peaks (66 and 122 kDa). Following calibration, 50 μL injections of HTNV-L apo and 5′ vRNA:HTNV-L were performed at several concentrations. For the preparation of 5′vRNA:HTNV-L sample, the 5′vRNA was incubated with HTNV-L at a 10:1 ratio in a buffer containing 30 mM HEPES pH 8, 250 mM NaCl, and 5 mM TCEP. SLS detection was performed online on an OMNISEC equipped with RALS and LALS detectors (Malvern) using a laser emitting at 660 nm. The protein concentration was measured online with the OMNISEC differential refractive index detector (Malvern) and a refractive index increment, dn/dc, of 0.191 mL.g⁻¹. The weight-average molar masses were calculated using the OMNISEC v5.10 software (Malvern).

### Mass spectrometry-based proteomic analyses

After separation by SDS-PAGE 6%, proteins were stained with Coomassie blue. The 250 kDa band (control) and the 150 kDa band of interest were cut out and contained proteins digested in-gel using trypsin (modified, sequencing purity, Promega), as previously described[32]. Two samples were therefore analyzed, and no replicate was done. The resulting peptides were analyzed by online nanoliquid chromatography coupled to MS/MS (Ultimate 3000 RSLCnano and Orbitrap Exploris 480, Thermo Fisher Scientific) using a 35 min gradient. For this purpose, the peptides were sampled on a precolumn (300 μm × 5 mm PepMap C18, Thermo Scientific) and separated in a 75 μm × 250 mm C18 column (Aurora Generation 3, 1.7 μm, IonOpticks). The MS and MS/MS data were acquired by Xcalibur (Thermo Fisher Scientific).

Peptides and proteins were identified by Mascot (version 2.8.0, Matrix Science) through concomitant searches against the UniProt database (Trichoplusia ni taxonomy, May 2023 version), a homemade database containing the sequence of HTNV-L, and a homemade database containing the sequences of classical contaminant proteins found in proteomic analyses (bovine albumin, keratins, trypsin, etc.). Trypsin/P was chosen as the enzyme, and two missed cleavages were allowed. Precursor and fragment mass error tolerances were set at respectively at 10 and 20 ppm. Peptide modifications allowed during the search were: carbamidomethyl (C, fixed), acetyl (protein N-term, variable), and oxidation (M, variable). The Proline software[33] was used for the compilation, grouping, and filtering of the results (conservation

of rank 1 peptides, peptide length ≥6 amino acids, false discovery rate of peptide-spectrum-match identifications <1%[34], and minimum of one specific peptide per identified protein group). Proline was then used to perform a compilation and grouping of the identified protein groups. Intensity-based absolute quantification (iBAQ) values[35] were calculated for each protein group in Proline using MS1 intensities of specific and razor peptides.

## Electron microscopy

To obtain the structures of HTNV-L apo symmetric dimer and hexamer, 3.5 μL of HTNV-L at 1 μM was deposited on glow-discharged (25 mA, 45 s) UltraAuFoil 300 mesh R1.2/1.3 grids (Quantifoil). The grid was blotted for 3 s (blot force 1) at 100% humidity and 4 °C in a Vitrobot Mark IV (Thermo Fisher Scientific) before plunge-freezing in liquid ethane.

To obtain the structure of 5′vRNA-bound HNTV-L monomers, HTNV-L at 1 μM was incubated with 10 μM 5′vRNA for 30 min at 4 °C. Grid preparation was identical to the one described for HTNV-L apo.

To obtain 3D structures of apo HTNV-L and 5′vRNA-bound HTNV-L, 14.650 and 26.745 micrographs were respectively collected on a 300 kV Titan Krios cryo-TEM microscope (Thermo Fisher Scientific) equipped with a K3 direct electron detector (Gatan) and a Gatan Quantum LS energy filter[36]. Coma and astigmatism correction were performed on a carbon grid. Automated multi-holes (4 × 4) data collection was performed with EPU. Movies containing 40 frames were acquired with a defocus ranging from −0.8 μm to −2.0 μm at a nominal magnification of 105.000x with a pixel size of 0.839 Å. The total exposure dose was 40 e⁻/Å$^2$.

## Image processing

To determine the structures of HTNV-L apo monomers/symmetric dimers/hexamers and HTNV-L 5′vRNA-bound monomers, micrograph movies were realigned using Motioncor2[37] (Supplementary Figs. 3, 4). The dose-weighted micrographs were imported into cryoSPARC 4.2.1[19] for CTF determination using the "Patch CTF estimation (multi)" tool. High-quality micrographs were selected with the "Manually Curate Exposure" tool and used for automated picking using the "Blob picker tool" utility. Different diameters of particles were used depending on the oligomeric state: between 90 and 190 Å for monomers, between 100 and 220 Å for dimers, and between 100 and 300 Å for hexamers.

For the determination of apo and 5′vRNA-bound HTNV-L monomeric structures, a second 2D classification was performed, followed by an ab initio reconstitution and a non-uniform refinement. For 5′vRNA-bound HTNV-L monomers, 3D classification without alignment with a global mask was used to separate the intermediate and the active conformations. Particles corresponding to each conformation were merged and used for a local refinement using a mask corresponding to the HTNV-L core.

For the determination of HTNV-L apo structures in symmetric dimers and in hexameric states, the 2D class averages obtained with the blob picker or their corresponding particles were used for template and Topaz picking[20]. All the particles picked with blob picker, Topaz, and template-picker that gave nice 2D class averages were used, following the removal of duplicates, to do an ab initio 3D reconstruction followed by a non-uniform refinement using the C2 symmetry. This provided global 3D reconstructions of the dimer and the hexamer.

To improve the density of the ENDO and the CBD of the dimer, symmetry expansion was done, followed by subtraction of the signal of one protomer. 3D classification without alignment focused either on the ENDO or the CBD provided one class in each case for which the ENDO or the CBD were more defined. A masked local refinement was then performed in each case.

Concerning the hexamer, to improve the density of the external dimer, a symmetry expansion was done, followed by signal subtraction of a central protomer and an external dimer. A 3D classification

without alignment separated particles with a better density of the external dimer. These particles were used for local refinement.

Local resolution was estimated for the maps of HTNV-L apo monomers/symmetric dimers/hexamers and HTNV-L 5′vRNA-bound monomers and used to filter them.

To compare symmetric apo dimer and 5′vRNA-related dimer, projections of the 3D cryo-EM map of the symmetric apo dimer masked around one protomer were compared to the 2D class of the 5′vRNA-related dimer masked around one protomer using Relion4.0.1[38] and eman2[39].

## Model building in the cryo-EM maps

For the refinement of HTNV-L apo and 5′vRNA-bound monomers, HTNV-L apo core (PDB: 8C4S) and HTNV-L 5′vRNA-bound cores (PDB: 8C4T) were rigidly fitted in Chimera[40]. In Coot 0.9.8.1[41], restraints at 4.3 Å generated with the restraint module were used for flexible refinement to fit the main chain into density. The missing regions were manually built into COOT. Careful validation of every side chain position was performed in COOT before refinement in real-space using Phenix[42].

For the refinement of HTNV-L apo isolated dimer, HTNV-L ENDO crystal structure (PDB: 5IZE), HTNV-L WT apo core (described in this article), and an AlphaFold model[43] of the mid-link, the CBD and the CTER domains of HTNV-L were rigidly fitted in Chimera[40]. The missing regions, in particular the lariat domain, the C-terminal region of the ENDO, and missing loops of the core region, were manually built into COOT. Careful validation of every side chain position was performed in COOT for every part except the CBD before being refined in real-space using Phenix[42].

For the refinement of HTNV-L hexamer, the HTNV-L apo dimer structure was split into different rigid parts, namely the ENDO, the core, the mid-link associated with the lariat α-helix, the CBD, and the CTER. These parts were used for rigid body fitting in the central dimer and in the most resolved external protomers. Careful validation of every side chain position was performed in COOT before being refined using Phenix real-space refinement[42]. For the less resolved external protomers, the map resolution was not sufficient for model building, and the protomers were thus separated into isolated core, ENDO, and C-terminal regions that were rigidly fitted in the focused map containing a central protomer and the external dimer. Interfaces between the less resolved external protomers and their interacting protomers were optimized in coot.

For each model, atomic model validation was performed with Molprobity[44] and the PDB validation server. Model resolution according to the cryo-EM maps was estimated with Phenix at the 0.5 FSC cutoff. Protein-protein interactions were analysed with PISA[27]. Figures were created using ChimeraX[45].

## Reporting summary

Further information on research design is available in the Nature Portfolio Reporting Summary linked to this article.

# Data availability

The coordinates and structure factors generated in this study have been deposited in the Protein Data Bank and the Electron Microscopy Data Bank database under accession codes: 8QE5 and EMD-18343 (HTNV-L apo monomer); 8QGT and EMD-18390 (5′vRNA-bound HTNV-L monomer in the intermediate state); 8GH3 and EMD-18397 (5′vRNA-bound HTNV-L monomer in the activated state); 8QGU and EMD-18391 (chimeric map of HTNV-L apo dimer associated with the complete model of HTNV-L dimer); EMD-18392 (cryo-EM map of HTNV-L apo dimer); EMD-18393 (cryo-EM map focused on the ENDO of a protomer from HTNV-L apo dimer); EMD-18394 (cryo-EM map focused on the CBD of a protomer from HTNV-L apo dimer); 8QHD and EMD-18408 (chimeric map of HTNV-L hexamer associated with the complete

model of HTNV-L hexamer); EMD-18406 (cryo-EM map of HTNV-L hexamer); EMD-18405 (cryo-EM map of HTNV-L hexamer focused on one central protomer and two external protomers). The cryo-EM raw data[46,47] are available under restricted access for a period of 3 years (ESRF embargo), access can be obtained before the end of this embargo by request to H.M. or after the end of this embargo through the ESRF's data portal. The mass spectrometry proteomics data have been deposited to the ProteomeXchange Consortium via the PRIDE[48] partner repository with the dataset identifier PXD049232.

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

## Acknowledgements

We thank Guy Schoehn for setting up and maintaining the IBS/ISBG EM platform and for discussion; Eleftherios Zarkadas for support and technical advices on cryo-EM data collection on the IBS/ISBG Thermo Fisher Glacios; Lindsay Mcgregor and Romain Linares for data collection on the ERSF CM01 Krios; Martin Pelosse for technical advices on expression; Caroline Mas for training on the mass photometer and SEC-SLS experiments; Yohann Couté for LC-MS/MS experiment and analysis; Aymeric Peuch for setting up and maintaining the EM computing cluster; Ambroise Desfosses for advices on cryo-EM image processing; Daphna Fenel and Madalen Le Gorrec for technical support. This work used the platforms of the Grenoble Instruct-ERIC center (ISBG; UAR 3518 CNRS-CEA-UGA-EMBL) within the Grenoble Partnership for Structural Biology (PSB) supported by FRISBI (ANR-10-INBS-05-02) and GRAL, financed within the University Grenoble Alpes graduate school (Ecoles Universitaires de Recherche) CBH-EUR-GS (ANR-17-EURE-0003). The electron microscope facility is supported by the Auvergne-Rhône-Alpes Region, the Fondation pour la Recherche Médicale (FRM), the fonds FEDER and the GIS-Infrastructures en Biologie Santé et Agronomie (IBiSA). We thank the platform staff who enabled us to perform these analyses. IBS acknowledges integration into the Interdisciplinary Research Institute of Grenoble (IRIG, CEA). This work was supported by the ANR grant ANR-19-CE11-0024, the Institut Universitaire de France endowment to H.M., and the Fondation pour la Recherche Médicale grant FDT202304016394 to Q.D.T.

## Author contributions

Q.D.T., B.A., and P.T. expressed and purified HTNV-L. Q.D.T. performed mass photometry and SEC-SLS measurements. H.M. and Q.D.T. collected cryo-EM data on a Thermo Fischer Scientific Glacios EM. Q.D.T. and B.A. performed cryo-EM image processing. B.A. determined the cryo-EM reconstruction of HTNV-L dimers. Q.D.T. determined all the deposited structures. H.M., B.A., and Q.D.T. built the models based on the cryo-EM maps. H.M., Q.D.T., D.H., and B.A. performed structural analysis. H.M. supervises Q.D.T., P.T., and B.A. The project was conceived and used funding obtained by H.M. The manuscript was written by H.M., Q.D.T., and D.H. with inputs from B.A.

## Competing interests

The authors declare no competing interests.
