## [Peer Review File · Nature Communications]

REVIEWER COMMENTS

Reviewer #1 (Remarks to the Author):

Using cryo-EM, Trouilleton and co-authors reported the structures of monomers, oligomers, and viral RNA complexes of complete HTNV-L protein that is the multi-functional polymerase containing an ENDO domain, the RdRp core, a Mid-link region with a CBD (cap-binding domain) inserted, and the C-terminal region (CTER) with an inserted lariat part. With the help of the particle picking technique, the authors resolve and analyze structures of monomers/protomers, dimers, and a hexamer.

The major discoveries they made include: first, 2 different conformations of HTNV L protomers with significant reconfigurations. Altered modes of domain-domain interactions with the RdRp core caused by this conformational change are disclosed; second, 2 forms of dimers enabled by the 2 conformations of protomers; and third, the formation of an HTNV-L hexamer by the 2 forms of dimers. The authors further showed that viral RNA binding induces structural change and results in oligomer disruption. Their 5' vRNA-bound HTNV-L monomer structure clearly showed the transformation of the active site motif E from an α -helix to a beta-sheet and supports the idea that RNA binding clashes with the dimerization of HTNV-L. In summary, this manuscript provides novel insights into the mechanism of HTNV polymerase in terms of oligomerization, which is very inspiring in the field of RNA virus RdRp research.

Overall, it is a paper with solid experimental data. The reported data unambiguously showed the conclusion. However, there are still some issues in this manuscript that need to be addressed:

1. Line 94, "5%" should be "2%" as shown in figure 1a.
2. Line 113, it may be better to rephrase not to use the 2 "central"
3. In line 188, it is 83 degrees, while in Fig 3a, it's 89, please correct.
4. In line 210, it is 2145, while in Fig 4b, it's 2142, please correct.
5. Line 217 not all the residues mentioned above are 100% conserved, for example, C2119.
6. Line 231, S1297, Y1543, and R1544 that are mentioned above are conserved.
7. Line 247, "fig. 3b" should be corrected to "Fig. 4b"
8. Line 252-253, residues 1301-1303 and 1252-1256 mentioned here are not shown in the figure or table.
9. Line 255, table 1 does not have related content, maybe "Fig. 5a" should be added here
10. Line 331, please rephrase "a common very important feature."
11. Line 366, reference 27 should be corrected to 29.

12. Line 367, reference 29 should be corrected to 28.
13. Line 445, the authors may explain what the meaning is of "3'vRNA1-25"
14. Line 494, add "on" after "based."
15. Line 504, from supplementary fig. 1, one cannot find related information, please correct.
16. Figure 6a, for the HTNV-L + 5'mut/3'vRNA sample the authors only collected ~1/5 of images that were collected for the apo and 5'vRNA complex, and also a different detector was used. The author may consider collecting more images to support the description in line 284 or explain the data collection difference in the body text.
17. Supplementary Fig. 3, caption: "Region used for masking are indicated with a dotted line" seems not applicable here.
18. Supplementary Fig. 11, the authors may consider enlarging the black arrows that show the structural change.

Reviewer #2 (Remarks to the Author):

Reviewer #3 (Remarks to the Author):

The scope of this review is limited primarily to analyzing the mass photometry results within the broader context of the study.

The manuscript by Trouillette et al. presents a cryo-electron microscopy study on the structure of Hantaan virus polymerase, describing its various oligomeric forms. The authors discovered that this polymerase can exist in two distinct conformations, leading to the formation of homodimers and hexamers that stabilize several protein domains. The binding of viral RNA induces conformational changes, leading to the disruption of the oligomers. These findings offer novel insights into the mechanism of Hantavirus polymerase multimerization and have the potential to enhance our understanding of its enzymatic function.

In their manuscript, the authors present a substantial amount of carefully collected experimental data. However, unfortunately, the biophysical characterization of the protein oligomerization in solution, which is crucial for the interpretation of the cryo-EM results, is inadequate. Consequently, it is necessary to provide additional data to offer context, significance, and validation to the structural findings.

At a minimum, the authors should provide a characterization of the oligomeric state in solution at the protein concentration utilized for cryo-EM sample preparation. This should include a concentration series to demonstrate the equilibrium behavior of the oligomer populations. It is essential to obtain these data both in the presence and absence of the vRNA. Ideally, determining the equilibrium constants for oligomer formation would be highly desirable.

Specific comments:

The authors should identify the species that is present at the approximate 150 kDa position in the MP distributions depicted in Figure 1a. The authors suggest that this species "could potentially correspond to the HTNV-L core only, too minoritarian to be detected in SDS-PAGE." However, even a 10% impurity should be detectable on SDS-PAGE, and Supplementary Figure 1c demonstrates that the population of this species reaches up to 30% in other SEC fractions.

Critical analysis of the data presented in Supplementary Figure 1 could provide valuable insights into the nature of this impurity. The similarity in the pattern of protein oligomerization across samples collected at different elution times (Figure S1c) suggests that the protein oligomer population reaches equilibrium at the concentration used for MP analysis. However, the early dilution fractions exhibit a higher population of the approximately 150 kDa component. These fractions also display an elevated 255 nm to 280 nm ratio in the chromatogram, indicating a potential presence of nucleic acid-containing complexes. This could explain their absence on the SDS-PAGE. Nevertheless, the presence of RNA-protein complexes is unlikely in material treated with ribonuclease, as stated in the materials and methods section.

Figure 6: It is peculiar that the authors did not supplement this data with an in-solution oligomerization study conducted in the presence of vRNA, considering that the disruption of the oligomers upon RNA binding is one of the key findings of this study.

Lines 308-311: The authors discuss the equilibrium between monomers, dimers, and hexamers, and estimate the "dissociation constant of the interaction is roughly in the micromolar range." However, there are several issues with this estimation. Firstly, the complex oligomerization process involving the formation of multiple oligomeric species (dimers, trimers, tetramers, and hexamers) cannot be accurately described by a single dissociation constant. Secondly, since all these oligomers are present in the MP distributions collected at a protein concentration of 30 nM, the oligomerization is likely much stronger than suggested by the micromolar estimate. Furthermore, the MP data were obtained immediately after a 40-fold dilution of the protein sample, making it challenging to draw conclusions without control experiments that assess the presence of kinetic effects on dissociation.

Reviewer #4 (Remarks to the Author):

Thank you for sending the manuscript by Quentin Durieux Trouillette et al. This manuscript describes the complete cryo-EM reconstruction of the Hantaan virus RNA-dependent RNA polymerase (HTNV-L or RdRP). While structures of the core as well as the ENDO domain have been available, we are now able to appreciate the structure of the entire protein. In particular, we can see the ENDO and Cap-binding domain in the context of the full-length protein (for the latter an alphafold model was docked into maps of lower local resolution). One of the most interesting findings is that HTNV-L adopts different oligomeric states although the physiological role, if any, of this multimers remains to be shown.

I have no objection to publication but would encourage the authors to take my comments into account:

Comments:

I think one of the most intriguing results are the different multimers and I would suggest the following experiments to corroborate them:

1.: The authors could repeat their size-exclusion/mass-photometry experiments at various ionic strengths to provide information on the stability of the multimers.

2.: I think the manuscript would benefit from experiments where the HTNV-L protein is incubated with viral RNA followed by size-exclusion/mass-photometry to confirm that multimers decrease as a function of RNA ligand in solution.

3.: Do the authors have an explanation for the ordering on motif F in the HTNV-L dimer as opposed to mono- and hexamer? The superposition in Fig. 2d suggests no changes in the environment of motif F so it is difficult to understand what causes this change.

4.: Line 219: Unclear to me what Supplementary Data 1 refers to. I suggest to color the interface of the CTER domains by conservation so readers can appreciate it more easily. This can be done by providing alignments to ChimeraX to color residue by their conservation.

Minor observations:

I have noticed a large number of typos and tried to list them here but I'm sure I missed most of them – please proof-read carefully.

Line 94 – should read species

Line 95 could potentially read “could correspond to HTNL-V core, a potential degradation product of the full-length protein but it too low in abundance to be detected by SDS-PAGE.”

Line 114 – 116: sentence not clear

Line 118: begins not begin

Line 357/358: should read “... become the major species, ...”

Line 381/382: should red “...can become the majority (or major species) ...”

Line 386: similar to previous comment

Dear Dr. Serrano, Dear Referees,

We thank you very much for your constructive comments on our manuscript entitled "Structural characterization of the oligomerization of full-length Hantaan virus polymerase into symmetric dimers and hexamers". We think they really helped us to improve our manuscript. Please find below our point-by-point responses.

Reviewer #1 (Remarks to the Author):

Using cryo-EM, Trouilleton and co-authors reported the structures of monomers, oligomers, and viral RNA complexes of complete HTNV-L protein that is the multi-functional polymerase containing an ENDO domain, the RdRp core, a Mid-link region with a CBD (cap-binding domain) inserted, and the C-terminal region (CTER) with an inserted lariat part. With the help of the particle picking technique, the authors resolve and analyze structures of monomers/protomers, dimers, and a hexamer.

The major discoveries they made include: first, 2 different conformations of HTNV L protomers with significant reconfigurations. Altered modes of domain-domain interactions with the RdRp core caused by this conformational change are disclosed; second, 2 forms of dimers enabled by the 2 conformations of protomers; and third, the formation of an HTNV-L hexamer by the 2 forms of dimers. The authors further showed that viral RNA binding induces structural change and results in oligomer disruption. Their 5' vRNA-bound HTNV-L monomer structure clearly showed the transformation of the active site motif E from an α -helix to a beta-sheet and supports the idea that RNA binding clashes with the dimerization of HTNV-L. In summary, this manuscript provides novel insights into the mechanism of HTNV polymerase in terms of oligomerization, which is very inspiring in the field of RNA virus RdRp research.

Overall, it is a paper with solid experimental data. The reported data unambiguously showed the conclusion. However, there are still some issues in this manuscript that need to be addressed:

1. Line 94, "5%" should be "2%" as shown in figure 1a.

Thank you for pointing this out. It has been corrected.

2. Line 113, it may be better to rephrase not to use the 2 "central"

This has been corrected. The new sentence is:

"The central dimer acts as an anchor on which two external dimers bind symmetrically."

3. In line 188, it is 83 degrees, while in Fig 3a, it's 89, please correct.

The correction has been done.

4. In line 210, it is 2145, while in Fig 4b, it's 2142, please correct.

This has been corrected.

5. Line 217 not all the residues mentioned above are 100% conserved, for example, C2119.

Reviewer 1 is correct, most of the residues are conserved but some are not. We have modified to:

“Importantly, most of the residues involved in the interactions between the CTER domains are conserved amongst Hantaviruses (Supplementary Fig. 12, Supplementary Data 1)”

We have added an additional **Supplementary Fig. 12** in which the residues involved in the interaction between the CTER domains are colored by conservation.

6. Line 231, S1297, Y1543, and R1544 that are mentioned above are conserved.

Reviewer 1 is correct. However, S1297 interacts with L1294 that is not conserved. Y1543 and R1544 interact with S1444 and Q1445, that are not conserved. The interaction is not conserved, and therefore we cannot mention any conservation in the text.

7. Line 247, “fig. 3b” should be corrected to “Fig. 4b”

Thank you. This has been corrected.

8. Line 252-253, residues 1301-1303 and 1252-1256 mentioned here are not shown in the figure or table.

The mentioned residues belong to the thumb domain. To keep **Fig. 5a** as clear as possible, we have colored the thumb domains of protomer B1 and B2 and we have simplified the text as follows:

“In addition, the polymerase cores of the two protomers B make hydrophobic contacts through their respective thumb domains, although their contribution to the dimer’s interface remain modest (19%) (Fig. 5a and Table 1)”.

9. Line 255, table 1 does not have related content, maybe “Fig. 5a” should be added here

We have modified the sentence to make it clearer:

“Finally, the lariat residues interact with the polymerase core of the other protomer (Table 1).”

There are two lines in **Table 1** that mention the interactions between:

- the lariat B1 and the core B2,
- the core B1 and the lariat B2.

10. Line 331, please rephrase “a common very important feature.”

We have rephrased as follows:

“The CTER domains display the largest variability amongst bunyaviruses, but, interestingly, they all contain a large protrusion that is named β -hairpin strut in LACV-L, and lariat in BDV-L and HTNV-L.”

11. Line 366, reference 27 should be corrected to 29.

12. Line 367, reference 29 should be corrected to 28.

Thank you for pointing out a problem in the numbering of some references. This has been corrected.

13. Line 445, the authors may explain what the meaning is of “3’vRNA1-25”

14. Line 494, add “on” after “based.”

15. Line 504, from supplementary fig. 1, one cannot find related information, please correct.

16. Figure 6a, for the HTNV-L + 5’mut/3’vRNA sample the authors only collected ~1/5 of images that were collected for the apo and 5’vRNA complex, and also a different detector was used. The author may consider collecting more images to support the description in line 284 or explain the data collection difference in the body text.

Comments 13 to 16 are all referring to HTNV-L in complex with 5' and 3'vRNA. Although the results we presented in the initial article were correct and the minor points 13 to 16 raised by Reviewer 1 could have addressed without problem, we propose here to remove from the article the discussion about the effect of 3'vRNA binding. It was presented as a control in the initial version of the article and was therefore not very detailed.

Indeed, following the comments of Reviewers 3 and 4, we have now expanded and reinforced the data on apo and 5'vRNA-bound HTNV-L. We think that the main message of the article is now to describe the structures of monomer, symmetric dimer and hexamer present in the apo state, analyze their equilibrium, and analyze the effect of the 5'vRNA in the modification of the structures and the equilibrium between species. To emphasize the message, we thus now present a detailed analysis of the apo and the 5'vRNA-bound states, and we have removed the 5'/3'vRNA data.

17. Supplementary Fig. 3, caption: "Region used for masking are indicated with a dotted line" seems not applicable here.

This has been corrected in **Supplementary Fig. 4** (new figure number due to the addition of new supplementary figures).

18. Supplementary Fig. 11, the authors may consider enlarging the black arrows that show the structural change.

This has been modified. Please note that **Supplementary Fig. 11** is now **Supplementary Fig. 13** as two additional Supplementary Figures have been added to follow Reviewers requests.

Reviewer #2 (Remarks to the Author):

I co-reviewed this manuscript with one of the reviewers who provided the listed reports. This is part of the Nature Communications initiative to facilitate training in peer review and to provide appropriate recognition for Early Career Researchers who co-review manuscripts. Thank you for this.

Reviewer #3 (Remarks to the Author):

The scope of this review is limited primarily to analyzing the mass photometry results within the broader context of the study.

The manuscript by Trouilleton et al. presents a cryo-electron microscopy study on the structure of Hantaan virus polymerase, describing its various oligomeric forms. The authors discovered that this polymerase can exist in two distinct conformations, leading to the formation of homodimers and hexamers that stabilize several protein domains. The binding of viral RNA induces conformational changes, leading to the disruption of the oligomers. These findings offer novel insights into the mechanism of Hantavirus polymerase multimerization and have the potential to enhance our understanding of its enzymatic function.

Thank you for this positive comment.

1. In their manuscript, the authors present a substantial amount of carefully collected experimental data. However, unfortunately, the biophysical characterization of the protein oligomerization in solution, which is crucial for the interpretation of the cryo-EM results, is inadequate. Consequently, it is necessary to provide additional data to offer context, significance, and validation to the structural findings.

Thank you for this constructive comment. Indeed, as mentioned by Reviewer 3, our data was mainly focused on the structural determination of the newly discovered oligomers formed by HTNV-L. We are happy to see the positive comments of all the Reviewers on the structural data and their analysis. We have now added a more detailed biophysical characterization of HTNV-L in solution by performing size-exclusion chromatography coupled to static light scattering (SEC-SLS) (concentration series, in presence and absence of 5'vRNA) and additional mass photometry assays (at different ionic strengths, in presence and absence of 5'vRNA).

2. At a minimum, the authors should provide a characterization of the oligomeric state in solution at the protein concentration utilized for cryo-EM sample preparation. This should include a concentration series to demonstrate the equilibrium behavior of the oligomer populations. It is essential to obtain these data both in the presence and absence of the vRNA. Ideally, determining the equilibrium constants for oligomer formation would be highly desirable.

We have performed SEC-SLS at concentrations ranging from 2.4 to 14 μM in the absence of vRNA. This range was chosen as the protein used for cryo-EM is injected at 8 μM on a SEC for the last purification step. The concentration series therefore covers conditions that are around the ones used for cryo-EM preparation. The results show a main peak that varies in mass depending on the HTNV-L concentration injected, that is a clear indication of an equilibrium that prevents the separation of the different species. For apo HTNV-L, we decided not to indicate any dissociation constants as several HTNV-L oligomers are identified by mass photometry, resulting in a complex signal that prevents the unambiguous determination of separate Kds.

The text has been updated as follows:

“Size-exclusion chromatography coupled to static light scattering (SEC-SLS) confirms this equilibrium with the detection of a single peak whose weight-averaged molar mass increases from 296 to 374 kDa upon HTNV-L concentration increase from 2.4 to 14 μM (Fig. 1b). The complex mixture present in this single peak unfortunately prevents the deduction of the multiple dissociation constants between the different species.”

Equivalent experiments were performed in presence of 5'vRNA with concentrations ranging from 1.36 to 6.37 μM (the lower concentration of the measurements in presence of 5'vRNA originates from the fact the concentrated HTNV-L sample is purified at high NaCl concentration to be stable and has to be diluted to 250 mM NaCl to ensure efficient 5'vRNA-binding prior to SEC-SLS analysis). An equilibrium is also visualized, and the text has been updated as follows:

“SEC-SLS of HTNV-L incubated with 5'vRNA results in a single peak with a weight-averaged molar mass that varies from 269 to 382 kDa when HTNV-L concentration is increased from 1.36 to 6.37 μM (Fig. 6b).”

Considering the monomer/5'vRNA-related dimer equilibrium only, it is in theory possible to calculate a K_d using the equations described in Benfield et al, JBC, 2011, DOI: 10.1074/jbc.M111.231381 :

$$M_w = M_M \left(\frac{8[M]_T + K_d - \sqrt{K_d^2 + 8K_d[M]_T}}{4[M]_T} \right)$$

M_w being the weight-average molar mass identified by SEC-SLS, M_M being the molecular mass of HTNV-L monomer and $[M]_T$ being the total concentration of HTNV-L measured by refractive index. However in our case, such a calculated K_d might be too inaccurate as (i) the 150kDa peak is only partially separated from the monomer-dimer equilibrium and (ii) the instability of HTNV-L prevents adding higher concentration of HTNV-L to reach the asymptote in the graph "weight-averages molar mass = f(concentration)" shown in an inset in **Fig. 6b**.

We have nevertheless performed the calculation but, for the reasons mentioned above, we decide not to include the calculated K_d in the main text. To answer Reviewer 3 as best, we just mention here that the estimated $K_{d_{\text{monomer}/5'v\text{RNA-related dimer}}}$ is in the 10 μM range.

3. Specific comments: The authors should identify the species that is present at the approximate 150 kDa position in the MP distributions depicted in Figure 1a. The authors suggest that this species "could potentially correspond to the HTNV-L core only, too minoritarian to be detected in SDS-PAGE." However, even a 10% impurity should be detectable on SDS-PAGE, and Supplementary Figure 1c demonstrates that the population of this species reaches up to 30% in other SEC fractions. Critical analysis of the data presented in Supplementary Figure 1 could provide valuable insights into the nature of this impurity. The similarity in the pattern of protein oligomerization across samples collected at different elution times (Figure S1c) suggests that the protein oligomer population reaches equilibrium at the concentration used for MP analysis. However, the early dilution fractions exhibit a higher population of the approximately 150 kDa component. These fractions also display an elevated 255 nm to 280 nm ratio in the chromatogram, indicating a potential presence of nucleic acid-containing complexes. This could explain their absence on the SDS-PAGE. Nevertheless, the presence of RNA-protein complexes is unlikely in material treated with ribonuclease, as stated in the materials and methods section.

Reviewer 3 suggests that the 150kDa visualized in mass photometry could correspond to nucleic-acid containing complexes. To test this idea, we incubated HTNV-L with either DNase, RNase or trypsin and performed mass photometry (**Rebuttal letter Figure 1**). We used for this purpose HTNV-L in a buffer containing 150 mM NaCl as this condition gives the highest percentage of 150kDa species in mass photometry. These tests reveal that DNase and RNase do not change the percentage of the 150kDa species detected. The mass photometry experiment done in presence of trypsin shows a low signal at molecular weight around 100kDa. As proteins that are 40 kDa or less cannot be visualized in the conditions used for mass photometry, this result suggests that trypsin degrades the protein. Altogether, these results suggest that the 150kDa signal is composed of protein.

Rebuttal letter Figure 1 Mass photometry of apo HTNV-L in presence of DNase, RNase or Trypsin

Mass photometry of HTNV-L in presence of DNase, RNase and Trypsin at 150mM NaCl. For each condition, the protein was incubated 2h at room temperature before measurement. The HTNV-L:DNase, RNase or Trypsin molar ratio is 1:0.1

To then identify this protein, we performed a bottom-up mass spectrometry (MS)-based proteomic analysis of a 150kDa band extracted from an SDS-PAGE gel. This approach identifies HTNV-L as the most abundant protein of this band, with detected peptides that originate from the entire sequence of HTNV-L. We thus draw the hypothesis that the 150kDa species may contain different cleaved forms of HTNV-L that co-migrate in this particular SDS-PAGE gel band.

The following paragraph and a **Supplementary Fig. 2** have been added:

“To characterize the proteins present in the 150kDa band, a bottom-up mass spectrometry (MS)-based proteomic analysis was performed. For this, proteins present in the 150 kDa band were in-gel digested with trypsin before analysis by nano liquid chromatography coupled to MS/MS. Obtained result revealed that the most abundant protein in this band was HTNV-L (Supplementary Fig. 2a). The identified peptides were covering the entire HTNV-L sequence (Supplementary Fig. 2b), suggesting that the 150kDa band may contain different cleaved forms of HTNV-L co-migrating in this particular SDS-PAGE gel band.”

4. Figure 6: It is peculiar that the authors did not supplement this data with an in-solution oligomerization study conducted in the presence of vRNA, considering that the disruption of the oligomers upon RNA binding is one of the key findings of this study.

We would like to really thank Reviewers 3 and 4 for this important point. We have now performed bio-physical analysis of HTNV-L in presence of 5'vRNA by mass-photometry and size-exclusion chromatography coupled by static light scattering. Both these methods detected dimers in presence of 5'vRNA. This was surprising as the cryo-EM analysis of the 5'vRNA:HTNV-L sample detected only a very small portion of symmetric dimers. This prompted us to reanalyze our 5'vRNA:HTNV-L cryo-EM data. We confirm here our initial

conclusion: only very few symmetric dimers are detected, confirming their disruption in presence of 5'vRNA. Reprocessing of the dataset however identified that 7.6 % of the particles correspond to another type of 5'vRNA-related dimer. Unfortunately, and despite the large amount of data collected (26.745 movies collected on a Titan Krios), this 5'vRNA-related dimer shows a strong preferential orientation on grids, currently preventing its 3D characterization. This precludes the analysis of (i) the conformations of the protomers and (ii) the interactions between the protomers in the 5'vRNA-related dimer. We therefore remain very cautious, we present these new results as preliminary, clearly stating that they open new perspectives which will be the scope of future research.

We present this finding in the text as follows:

-at the end of the introduction:

“Our results finally reveal that incubation of HTNV-L with 5'vRNA triggers large conformational changes in the protomers and leads to a new equilibrium between active monomers and yet-to-be characterized 5'vRNA-related dimers which are different from the apo dimers.”

-in the results, paragraph: ***Incubation with 5'viral RNA disrupts HTNV-L apo multimers and results in the formation of another type of HTNV-L dimers***

“To analyze if incubation with 5'vRNA is modifying HTNV-L conformation and oligomerization mode, a cryo-EM dataset was collected in presence of 5'vRNA. Intriguingly, cryo-EM 2D class averages reveal that only 0.3% of symmetric dimers and 0.2% of symmetric hexamers are detected in presence of 5'vRNA, the extremely low number of particles preventing further structural characterization (Fig. 6c). This strongly differs from the 2D class averages visualized in the absence of vRNA in which 10.1 % and 2.1 % of the particles correspond to symmetric dimers and hexamers (Fig. 6c). We could speculate that the symmetric multimers detected in the dataset collected with 5'vRNA could correspond to the few particles that would have remained in their apo form. Interestingly, another type of dimer is detected in 2D class averages of the HTNV-L dataset collected in presence of 5'vRNA (Fig. 6c). We will call this dimer “5'vRNA-related HTNV-L dimer”. When one polymerase core from the apo symmetric dimer is shown in the same orientation as one polymerase core of the 5'vRNA-related HTNV-L dimer, it clearly appears that the orientation of their second core differs (Fig. 6d). This, combined with the visualization of extra densities that may correspond to the ENDO and/or the C-terminal regions, suggests that the 5'vRNA-related HTNV-L dimer could be asymmetric (Fig. 6e). The dimers detected in presence of 5'vRNA represent 7.6% of the particles, a percentage roughly comparable to the 10.1% of symmetric dimers visualized in absence of vRNA, explaining why these two different types of dimers could not be distinguished by mass photometry and SEC-SLS due to their similar masses. Unfortunately, the 5'vRNA-related HTNV-L dimers adopt a strong preferential orientation on the cryo-EM grid preventing their 3D characterization. These preliminary results will need to be confirmed by future work that should solve the strong preferential orientation problem and determine the high-resolution 3D structure of the 5'vRNA-related HTNV-L dimers.”

In the discussion:

“In addition, we here observe that incubation with 5'vRNA results in the formation of another type of 5'vRNA-related HTNV-L dimer. The identification of such dimer is intriguing and opens the way towards future investigations that will decipher its exact 3D structure, the

conformational changes necessary for its formation and its functional role in replication and transcription."

The added panels **Fig. 6c-e** show a comparison between apo dimer and 5'vRNA-related dimer.

5. Lines 308-311: The authors discuss the equilibrium between monomers, dimers, and hexamers, and estimate the "dissociation constant of the interaction is roughly in the micromolar range." However, there are several issues with this estimation. Firstly, the complex oligomerization process involving the formation of multiple oligomeric species (dimers, trimers, tetramers, and hexamers) cannot be accurately described by a single dissociation constant. Secondly, since all these oligomers are present in the MP distributions collected at a protein concentration of 30 nM, the oligomerization is likely much stronger than suggested by the micromolar estimate. Furthermore, the MP data were obtained immediately after a 40-fold dilution of the protein sample, making it challenging to draw conclusions without control experiments that assess the presence of kinetic effects on dissociation.

We would like to thank Reviewer 3 for this remark. We have now performed SEC-SLS of HTNV-L at several concentrations ranging from 2.4 to 14 μ M. We visualize an equilibrium that prevents separation of the monomers, dimers, trimers, tetramers and hexamers. As a result, and in accordance with the comment of Reviewer 3, we don't provide a Kd estimation anymore. We have removed the sentence "dissociation constant of the interaction is roughly in the micromolar range" and instead mention:

"The complex mixture present in this single peak unfortunately prevents the deduction of the multiple dissociation constants between the different species."

Reviewer #4 (Remarks to the Author):

Thank you for sending the manuscript by Quentin Durieux Trouillette et al. This manuscript describes the complete cryo-EM reconstruction of the Hantaan virus RNA-dependent RNA polymerase (HTNV-L or RdRP). While structures of the core as well as the ENDO domain have been available, we are now able to appreciate the structure of the entire protein. In particular, we can see the ENDO and Cap-binding domain in the context of the full-length protein (for the latter an alphafold model was docked into maps of lower local resolution). One of the most interesting findings is that HTNV-L adopts different oligomeric states although the physiological role, if any, of this multimers remains to be shown.

We would like to thank Reviewer 4 for this positive evaluation of our manuscript.

I have no objection to publication but would encourage the authors to take my comments into account:

I think one of the most intriguing results are the different multimers and I would suggest the following experiments to corroborate them:

1.: The authors could repeat their size-exclusion/mass-photometry experiments at various ionic strengths to provide information on the stability of the multimers.

This has now been added on **Supplementary Fig. 1c** with NaCl concentration varying from 150 mM to 1M. We now describe the stability of the multimers as follows:

“In buffers containing 250mM NaCl, several species were automatically detected including monomers (53%), dimers (17%), higher molecular weight oligomers up to hexamers (2%) and an unidentified 150kDa protein (18%) (Fig. 1a). Increasing the ionic strength to 1M NaCl maintained the proportion of monomers/dimers, while no automatic detection was obtained for the oligomers ranging from trimers to hexamers (Supplementary Fig. 1c). Inversely, in a buffer containing 150mM NaCl, a large proportion of the 150kDa species were visualized (45%), while less monomers and dimers were present and higher oligomers were not detected (Supplementary Fig. 1c).”

The 150kDa species was identified as being HTNV-L by LC-MS/MS and probably corresponds to different cleaved forms of HTNV-L.

2.: I think the manuscript would benefit from experiments where the HTNV-L protein is incubated with viral RNA followed by size-exclusion/mass-photometry to confirm that multimers decrease as a function of RNA ligand in solution.

We would like to really thank Reviewer 4 for this very important point. We have performed mass-photometry not only at 250mM NaCl as requested, but also at different ionic strengths: *“To analyze if incubation with 5'vRNA is modifying the equilibrium between the different oligomers, mass photometry was performed in buffers containing different ionic concentrations (Fig. 6a, Supplementary Fig. 14). The results are very similar to the ones obtained in the absence of 5'vRNA except for the hexamers that could not be automatically detected. The 150kDa species are also less abundant, suggesting that the 5'vRNA stabilizes HTNV-L, partially preventing its degradation.”*

The visualization of dimers in presence of 5'vRNA was unexpected, so we have complemented the mass-photometry results with size-exclusion chromatography coupled with static-light scattering (SEC-SLS) to analyze the equilibrium between monomers and dimers:

“SEC-SLS of HTNV-L incubated with 5'vRNA results in a single peak with a weight-averaged molar mass that varies from 269 to 382 kDa when HTNV-L concentration is increased from 1.36 to 6.37 μ M (Fig. 6b).”

These results led us to re-process the cryo-EM data collected in presence of 5'vRNA and we identified that 7.6% of the data correspond to another type of dimer that is clearly different from the symmetric dimer observed in the apo form. This is now shown in the updated **Fig. 6c-e**. Unfortunately, a strong preferential orientation prevents its 3D structural characterization. This precludes the analysis of (i) the conformations of the protomers and (ii) the interactions between the protomers in this 5'vRNA-related dimer. We therefore remain very cautious, we present these new results as preliminary, clearly stating that they open new perspectives which will be the scope of future research.

The result section has been updated as follows:

“To analyze if incubation with 5'vRNA is modifying HTNV-L conformation and oligomerization mode, a cryo-EM dataset was collected in presence of 5'vRNA. Intriguingly, cryo-EM 2D class averages reveal that only 0.3% of symmetric dimers and 0.2% of symmetric hexamers are detected in presence of 5'vRNA, the extremely low number of particles preventing further

structural characterization (**Fig. 6c**). This strongly differs from the 2D class averages visualized in the absence of vRNA in which 10.1 % and 2.1 % of the particles correspond to symmetric dimers and hexamers (**Fig. 6c**). We could speculate that the symmetric multimers detected in the dataset collected with 5'vRNA could correspond to the few particles that would have remained in their apo form. Interestingly, another type of dimer is detected in 2D class averages of the HTNV-L dataset collected in presence of 5'vRNA (**Fig. 6c**). We will call this dimer "5'vRNA-related HTNV-L dimer". When one polymerase core from the apo symmetric dimer is shown in the same orientation as one polymerase core of the 5'vRNA-related HTNV-L dimer, it clearly appears that the orientation of their second core differs (**Fig. 6d**). This, combined with the visualization of extra densities that may correspond to the ENDO and/or the C-terminal regions, suggests that the 5'vRNA-related HTNV-L dimer could be asymmetric (**Fig. 6e**). The dimers detected in presence of 5'vRNA represent 7.6% of the particles, a percentage roughly comparable to the 10.1% of symmetric dimers visualized in absence of vRNA, explaining why these two different types of dimers could not be distinguished by mass photometry and SEC-SLS due to their similar masses. Unfortunately, the 5'vRNA-related HTNV-L dimers adopt a strong preferential orientation on the cryo-EM grid preventing their 3D characterization. These preliminary results will need to be confirmed by future work that should solve the strong preferential orientation problem and determine the high-resolution 3D structure of the 5'vRNA-related HTNV-L dimers."

The discussion has also been updated as follows:

"In addition, we here observe that incubation with 5'vRNA results in the formation of another type of 5'vRNA-related HTNV-L dimer. The identification of such dimer is intriguing and opens the way towards future investigations that will decipher its exact 3D structure, the conformational changes necessary for its formation and its functional role in replication and transcription."

This point was also raised by Reviewer 3 in his/her point 4. Reviewer 4 may therefore be interested in the related answer.

3.: Do the authors have an explanation for the ordering on motif F in the HTNV-L dimer as opposed to mono- and hexamer? The superposition in Fig. 2d suggests no changes in the environment of motif F so it is difficult to understand what causes this change.

Thank you for this point. We have carefully looked again at the cryo-EM maps of the monomer, dimer and hexamer. We confirm that the density for motif F in the monomer and hexamer is not defined enough to unambiguously build the amino acid sequence of motif F. Some blurry density is however present and is closing the template entry tunnel. For the dimer, the density for motif F is a bit clearer, enabling us to build a model, although with low precision due the flexibility. No clear change in the environment is visible between the dimer and monomer/hexamer. Reviewer 4 point is interesting and important as we should not oppose (i) dimer with an ordered motif F to (ii) monomer/hexamer with disordered motif F. The motif F is rather flexible in all cases.

We have thus updated the text as follows:

"The motif F is rather flexible and is positioned in the template entry tunnel, thereby closing it and confirming the inactive configuration of HTNV-L apo."

4.: Line 219: Unclear to me what Supplementary Data 1 refers to. I suggest to color the interface of the CTER domains by conservation so readers can appreciate it more easily. This can be done by providing alignments to ChimeraX to color residue by their conservation.

The **Supplementary Fig. 12** has been added and displays the interface of the CTER domains by conservation as proposed. The **Supplementary Data 1** complements the **Supplementary Fig. 12**. Indeed, it corresponds to a multiple alignment of full-length polymerases from different Hantaviruses.

Minor observations:

I have noticed a large number of typos and tried to list them here but I'm sure I missed most of them – please proof-read carefully.

We have carefully read the manuscript to remove as many typos as possible.

Line 94 – should read species

This has been corrected.

Line 95 could potentially read “could correspond to HTNL-V core, a potential degradation product of the full-length protein but it too low in abundance to be detected by SDS-PAGE.”

This has been updated following the nano-LC MS/MS experiments performed to identify this band.

Line 114 – 116: sentence not clear

We have rephrased the sentence:

“While the central dimer displays a unique configuration, the two external dimers adopt the same conformation as the isolated dimer.”

Line 118: begins not begin

This has been corrected.

Line 357/358: should read “... become the major species, ...”

Line 381/382: should read “...can become the majority (or major species) ...”

Line 386: similar to previous comment

This has been updated.

REVIEWERS' COMMENTS

Reviewer #1 (Remarks to the Author):

This revised submission has successfully addressed the concerns in the previous review.

Reviewer #2 (Remarks to the Author):

In this revised version of manuscript, the authors have optimized their focuses to stress on the oligomerization of the HTNV-L and the equilibrium of the monomer, dimers, and hexamer. The authors also concentrated on the 5'vRNA in the alteration of the structures and equilibrium to make the article better. I recommend its publication if my minor comments listed below would be addressed:

1. line 291, "dimer A" and "dimer B" seems to be mistakenly swapped: two dimer A and only one dimer B are in the hexamer;
2. line 294, in table 1, the number is "1351" while here it is "1310", please double check;
3. line 299 - 301, it should better to specify "5'vRNA is added"
4. line 443, add "of" before "these multimers"
5. line 801, here "180 degree" is used, in figure 3, "90 degree" should be corrected to "180 degree"
6. line 851, in figure 6b, "the crosses" may be indicated;
7. table 1, the 3 tables were not subtitled with "a, b, c"

Reviewer #3 (Remarks to the Author):

In their resubmitted manuscript, Trouillet and co-authors have addressed the previous concerns regarding insufficient biophysical characterization of the HTNV-L solution forms. They have now included additional mass photometry and size-exclusion chromatography/static light scattering data that provide important insights into the oligomeric behavior of the protein. Notably, data obtained for HTNV-L incubated with 5' vRNA led to the identification of another form of the HTNV-L dimer. The inclusion of the new data and the analysis the authors have now performed substantially improve the overall impact of the study. It is also a good illustration of the synergy between different biophysical methods and the importance of supporting structural data with solution experiments whenever possible. I have no further objections to publication.

Reviewer #4 (Remarks to the Author):

The reviewers have addressed all my concerns raised in the original review and I haven further objections.

Dear Dr. Serrano, Dear Referees,

We thank you very much for your comments on our revised manuscript entitled "Structural characterization of the oligomerization of full-length Hantaan virus polymerase into symmetric dimers and hexamers". We are delighted to see that we addressed all the previous concerns of Reviewer 1,3 and 4. We would like to thank Reviewer 2 for the careful check of our revised manuscript, we have corrected the 7 minor points listed.

Yours sincerely,

Hélène Malet

Reviewer #1 (Remarks to the Author):

This revised submission has successfully addressed the concerns in the previous review.

Reviewer #2 (Remarks to the Author):

In this revised version of manuscript, the authors have optimized their focuses to stress on the oligomerization of the HTNV-L and the equilibrium of the monomer, dimers, and hexamer. The authors also concentrated on the 5'vRNA in the alteration of the structures and equilibrium to make the article better. I recommend its publication if my minor comments listed below would be addressed:

1. line 291, "dimer A" and "dimer B" seems to be mistakenly swapped: two dimer A and only one dimer B are in the hexamer;
2. line 294, in table 1, the number is "1351" while here it is "1310", please double check;
3. line 299 - 301, it should better to specify "5'vRNA is added"
4. line 443, add "of" before "these multimers"
5. line 801, here "180 degree" is used, in figure 3, "90 degree" should be corrected to "180 degree"
6. line 851, in figure 6b, "the crosses" may be indicated;
7. table 1, the 3 tables were not subtitled with "a, b, c"

Reviewer #3 (Remarks to the Author):

In their resubmitted manuscript, Trouillet and co-authors have addressed the previous concerns regarding insufficient biophysical characterization of the HTNV-L solution forms. They have now included additional mass photometry and size-exclusion chromatography/static light scattering data that provide important insights into the oligomeric behavior of the protein. Notably, data obtained for HTNV-L incubated with 5' vRNA led to the identification of another form of the HTNV-L dimer. The inclusion of the new data and the analysis the authors have now performed substantially improve the overall impact of the study. It is also a good illustration of the synergy between different biophysical methods and the importance of supporting structural data with solution experiments whenever possible. I have no further

objections to publication.

Reviewer #4 (Remarks to the Author):

The reviewers have addressed all my concerns raised in the original review and I haven further objections.